# The Noise Geometry of Stochastic Gradient Descent

## Abstract

In this paper, we present a comprehensive analysis of the heterogeneous structure of minibatch noise, focusing on its favorable *alignment* with the landscape's local geometry (Wu et al., 2022). Specifically, we propose two metrics, derived from analyzing the influence of the noise structure on the loss and subspace projection dynamics separately, to quantify the alignment property. To showcase the practical relevance of our noise geometry characterization, we revisit the convergence analysis of stochastic gradient descent (SGD), revealing that the favorable noise geometry is crucial for ensuring benign convergence of SGD in high-dimensional settings. We also examine the noise geometry's influence on how SGD escapes from sharp minima. It is demonstrated that, unlike gradient descent (GD), which escapes sharp regions along the sharpest directions, SGD tends to escape through flatter directions. To support our theoretical findings, both synthetic and real-dataset experiments are provided.

## 1 Introduction

Minibatch SGD, along with its variants, now stands as the de facto optimizers in modern machine learning (ML) (Goodfellow et al., 2016). Unlike full-batch GD, SGD uses only mini-batches of data in each iteration, which injects noise into the training process. Considering a loss objective $\mathcal{L} : \mathbb{R}^p \mapsto \mathbb{R}$, the SGD iteration with learning rate $\eta$ can be written as

$$\theta_{t+1} = \theta_t - \eta \left( \nabla \mathcal{L}(\theta_t) + \xi_t \right), \tag{1}$$

where $\xi_t = \xi(\theta_t)$ denotes the injected noise that satisfies $\mathbb{E}[\xi_t] = 0$ and $\mathbb{E}[\xi_t \xi_t^\top] = \Sigma(\theta_t)$, with $\Sigma(\cdot)$ denoting the covariance matrix of SGD noise. Without loss of generality, throughout this paper, we assume that the batch size $B = 1$. The injected noise can significantly alter the optimizer's properties, particularly in aspects like optimization convergence (Hazan et al., 2016; Thomas et al., 2020; Wojtowytsch, 2023) and implicit regularization (Zhang et al., 2017; Keskar et al., 2017; Wu et al., 2017; 2018).

To illustrate this, we begin by examining the noise's influence on loss dynamics. Assuming $\mathcal{L}(\cdot)$ is twice differentiable and letting $H(\theta) = \nabla^2 \mathcal{L}(\theta)$, then the one-step loss update of SGD is given by

$$\mathbb{E}[\mathcal{L}(\theta_{t+1})] = \mathbb{E}[\mathcal{Q}(\theta_t)] + \frac{\eta^2}{2} \mathbb{E}[\gamma(\theta_t)] + o(\eta^2), \tag{2}$$

where $\mathcal{Q}(\theta) = \mathcal{L}(\theta) - \eta \|\nabla \mathcal{L}(\theta)\|^2 + \frac{\eta^2}{2} \nabla \mathcal{L}(\theta)^\top H(\theta) \nabla \mathcal{L}(\theta)$ denotes the contribution of GD component and $\gamma(\theta_t) = \mathbb{E}[\xi_t^\top H(\theta) \xi_t]$ represents the noise contribution. In optimization literature, one often makes the *bounded variance assumption*: $\mathbb{E}[\|\xi_t\|^2] \leq \sigma^2$, with $\sigma$ being a fixed constant. By additionally assuming $\|H(\theta_t)\|_2 \leq L$, it follows that $\gamma(\theta_t) \leq L\mathbb{E}[\|\xi_t\|^2] \leq L\sigma^2$. However, this estimates could substantially overestimate $\gamma(\theta_t)$ as it overlooks the fact: for typical ML models, both the magnitude and shape of SGD noise are state-dependent.

**Noise magnitude.** Bottou et al. (2018) introduced the affine variance assumption: $\mathbb{E}[\|\xi_t\|^2] \leq \sigma_0^2 + \sigma_1^2 \|\nabla \mathcal{L}(\theta_t)\|^2$, incorporating the state-dependence of noise magnitude. This assumption has been adopted in Faw et al. (2022) to justify the superiority of adaptive stochastic gradient methods. Additionally, a series of studies including Feng & Tu (2021); Mori et al. (2022); Wojtowytsch (2021); Liu et al. (2021) have shown, for regression with square loss, the magnitude of SGD noise can be bounded by the loss value: $\mathbb{E}[\|\xi_t\|^2] \leq \sigma^2 \mathcal{L}(\theta_t)$. This reveals that the noise magnitude diminishes to zero at global minima. Leveraging this property, some

works (Bassily et al., 2018; Fang et al., 2020; Wojtowytsch, 2021; Liu et al., 2023) showed that SGD can achieve convergence with a constant learning rate in the interpolation regime. This contrasts starkly with the bounded variance assumption, where SGD requires a decaying learning rate to converge (Hazan et al., 2016).

**Noise shape.** It is important to note that the noise contribution in (2) can be expressed as

$$\gamma(\theta_t) = \mathbb{E}[\xi_t^\top H(\theta_t)\xi_t] = \text{tr}(\Sigma(\theta_t)H(\theta_t)). \tag{3}$$

This underscores the significance of the noise shape, highlighting the necessity of examining the relationship between the noise covariance and the local Hessian. In particular, Zhu et al. (2019); Wu et al. (2020); Xie et al. (2020) have empirically demonstrated that $\Sigma(\theta)$ is highly *anisotropic* and bears resemblance to $H(\theta)$ to a certain degree for various ML models. However, it is still unclear how to quantitatively characterize this resemblance.

Beyond loss convergence, it has been demonstrated that the aforementioned noise geometry is also crucial in shaping the superior generalization properties of SGD (Zhu et al., 2019; Wu et al., 2020; Li et al., 2021; Wu et al., 2022; HaoChen et al., 2021) and general dynamical behaviors (Feng & Tu, 2021; Ziyin et al., 2022; Wu et al., 2018; Thomas et al., 2020). Therefore, it is important to establish quantitative descriptions of the above noise geometry.

In the context of regression with square loss, by incorporating the state dependence of both noise magnitude and shape, Mori et al. (2022) proposed a heuristic approximation of the noise covariance:

$$\Sigma(\theta) \approx 2\mathcal{L}(\theta)G(\theta), \tag{4}$$

where $G(\theta)$ denotes the empirical Fisher matrix and $G(\theta) \approx H(\theta)$ in low-loss region (see Section 2 for details). This approximation suggests an *intriguing alignment between SGD noise and local landscape*: 1) the noise magnitude is proportional to the loss value; 2) the noise energy tends to concentrate more along sharp directions than flat directions. The latter point can be deduced as follows: for a fixed direction $v$, the noise energy along $v$ satisfies

$$\mathbb{E}[(\xi(\theta)^\top v)^2] = v^\top \Sigma(\theta)v \approx 2\mathcal{L}(\theta)v^\top G(\theta)v, \tag{5}$$

where the second step uses (4) and $v^\top G(\theta)v$ is roughly the curvature of local landscape along the direction $v$. Unfortunately, it should be noted that this heuristic approximation is neither theoretically justified nor empirically verified.

## 1.1 Our contribution

The goal of this work is to advance beyond heuristic approximations of SGD noise by providing theoretical and quantitative characterizations. We highlight that the inaccuracy of approximation (4) stems from its attempt to establish a full description of the entire noise covariance. This is often unnecessary since, in most scenarios, our focus is only on some specific low-dimensional quantities instead of the entire high-dimensional trajectory. For instance, when analyzing the loss dynamics, it suffices to characterize $\gamma(\theta_t) \in \mathbb{R}$ as per Eq. (2) rather than $\Sigma(\theta_t) \in \mathbb{R}^{p \times p}$.

Adopting this perspective, we demonstrate that the alignment implied by approximation (4), while not accurate, is still valid when assessed using less stringent metrics. In addition, we present illustrative examples, showcasing the utility of our noise geometry quantification in analyzing SGD's dynamical behavior. Specifically, our key findings are outlined as follows.

- We first quantify the noise geometry through examining its influence on loss dynamics. According to Eq. (3), when approximation (4) holds, $\gamma(\theta) \approx 2\mathcal{L}(\theta)\|G(\theta)\|_F^2 =: \bar{\gamma}(\theta)$ in regions of low loss. This motivates us to measure the alignment strength using the ratio $\mu(\theta) = \gamma(\theta)/\bar{\gamma}(\theta)$. We prove that, in both (over-parameterized) linear models and two-layer nonlinear networks without bias, $\mu(\theta)$ is close to 1 across the entire parameter space, a result that intriguingly holds true regardless of the degree of over-parameterization.

- We next focus on the alignment along fixed directions, investigating if *the noise energy along a given direction is proportional to the curvature in that direction.* This type of alignment can be useful in describing the SGD dynamics in subspaces, such as the top principal components (see Section 4 for details). According to Eq. (5), we define the metric $g(\theta, v) = v^\top \Sigma(\theta) v / (2\mathcal{L}(\theta) v^\top G(\theta) v)$ to quantify the strength of directional alignment, where $v$ denotes the direction of interest. We establish that for (over-parameterized) linear models, this directional alignment also holds regardless of the degree of over-parameterization.

- Lastly, leveraging our noise characterization, we revisit the existing analyses of SGD's convergence and escape behaviors. Specifically, we obtain an improved convergence rate for SGD in high-dimensional linear regression, where the improvement stems from prior analyses overlooking the structured nature of SGD noise. In terms of escaping sharp minima, we show that the escape direction exhibits significant components along flatter directions, which contrasts sharply with GD, which escapes only along the steepest direction. Furthermore, we discuss an important implication of this escape property: cyclical learning rates can exploit this characteristic to help SGD locate flatter minima more effectively.

To validate our theoretical findings, we have provided both small-scale and large-scale experiments, including the classification of CIFAR-10 dataset using VGG nets and ResNets. Overall, we not only establish quantitative descriptions of noise geometry for SGD but also shed light on how the unique noise geometry helps SGD navigate the loss landscape.

## 1.2 Other related work

**The noise geometry.** Ziyin et al. (2022) conducted an analysis of the noise structure of online SGD for under-parameterized linear regression. Pesme et al. (2021) studied the implicit bias of SGD noise for diagonal linear networks. HaoChen et al. (2021); Damian et al. (2021); Li et al. (2022) showed that the noise covariance of label-noise SGD is $\varepsilon^2 G(\theta)$, where $\varepsilon$ denotes the size of label noise. Works such as Simsekli et al. (2019); Zhou et al. (2020) argued that the magnitude of SGD noise is heavy-tailed but not considered the noise shape. Lastly, we remark that aligning with our first part analysis, Wu et al. (2022) developed theoretical underpinnings for the noise geometry in terms of its contribution in loss dynamics. However, their analysis is restricted to OLMs under the infinite-data regime. In contrast, we offer an investigation of the finite-sample effects and extend substantially beyond the scope of Wu et al. (2022).

**Escape from minima and saddle points.** The phenomenon of SGD escaping from sharp minima exponentially fast was initially studied in Zhu et al. (2019); Wu et al. (2018). This provides an explanation of the famous "flat minima hypothesis" (Hochreiter & Schmidhuber, 1997; Keskar et al., 2017; Wu & Su, 2023)—one of the most important observations in explaining the implicit regularization of SGD. However, existing analyses of the escape phenomenon have primarily focused on the escape rate (Wu et al., 2018; Zhu et al., 2019; Xie et al., 2020; Mori et al., 2022; Ziyin et al., 2022). In contrast, we extends this focus by providing an analysis of escape direction, which is enabled by our refined description of the noise geometry. Kleinberg et al. (2018) introduced an alternative perspective, positing that SGD circumvents local minima by navigating an effective loss landscape that results from the convolution of the original landscape with SGD noise. In this context, our noise geometry results can be beneficial in understanding the effective loss landscape. In addition, prior works like (Daneshmand et al., 2018; Xie et al., 2022) has illustrated that the alignment of SGD noise with local geometry facilitates the rapid saddle-point escape of SGD. Our work offers theoretical support for the alignment assumptions in these studies.

## 2 Preliminaries

**Notation.** We use $\langle \cdot, \cdot \rangle$ for the Euclidean inner product and $\|\cdot\|$ the $\ell^2$ norm of a vector or the spectral norm of a matrix. For any positive integer $k$, let $[k] = \{1, \cdots, k\}$. Denote by $\mathcal{N}(\mu, S)$ the Gaussian distribution with mean $\mu$ and covariance matrix $S$. When $A$ is positive semidefinite, we use $\mathrm{srk}(A) := \mathrm{tr}(A)/\|A\|$ to denote the effective rank of $A$. We use $a \lesssim b$ to mean there exist an an absolute constant $C > 0$ such that $a \leq Cb$ and $a \gtrsim b$ is defined analogously. We write $a \sim b$ if both $a \lesssim b$ and $a \gtrsim b$ hold.

**Problem Setup.** Let $\{(x_i, y_i)\}_{i=1}^n$ with $x_i \in \mathbb{R}^d$ and $y_i \in \mathbb{R}$ for $i \in [n]$ be the training set and $f(\cdot; \theta) : \mathbb{R}^d \to \mathbb{R}$ be the model parameterized by $\theta \in \mathbb{R}^p$. Let $\ell_i(\theta) = \frac{1}{2}(f(x_i; \theta) - y_i)^2$ and $\mathcal{L}(\theta) = \frac{1}{n} \sum_{i=1}^n \ell_i(\theta)$ be the empirical loss. In theoretical analysis, we make the following input assumption:

**Assumption 2.1.** Suppose that $x_1, x_2, \ldots, x_n$ are $i.i.d$ samples drawn from $\mathcal{N}(0, S)$ with $S \in \mathbb{R}^{d \times d}$ denoting the input covariance matrix. We use $d_{\text{eff}} := \min\{\text{srk}(S), \text{srk}(S^2)\}$ to denote the effective input dimension.

In the above setup, the Hessian matrix of the empirical loss is given by

$$H(\theta) = G(\theta) + \frac{1}{n} \sum_{i=1}^n (f(x_i; \theta) - y_i) \nabla^2 f(x_i; \theta), \tag{6}$$

where $G(\theta) = \frac{1}{n} \sum_{i=1}^n \nabla f(x_i; \theta) \nabla f(x_i; \theta)^\top$ is the empirical Fisher matrix. Eq. (6) implies that when the fit errors are negligible, we have $G(\theta) \approx H(\theta)$ and in particular, at global minima $\theta^*$, $H(\theta^*) = G(\theta^*)$. Additionally, for linear regression $f(x; \theta) = \theta^\top x$, $H(\theta) = G(\theta) \equiv \frac{1}{n} \sum_{i=1}^n x_i x_i^\top$. Therefore, in this paper, we focus on examining the alignment between noise covariance $\Sigma(\theta)$ and the empirical Fisher matrix $G(\theta)$, rather directly analyzing the Hessian.

**SGD and noise covariance.** To minimize $\mathcal{L}(\cdot)$, SGD with a batch size of 1 updates as $\theta_{t+1} = \theta_t - \eta \nabla \ell_{i_t}(\theta_t)$ with $(i_t)_{t \geq 1}$ being $i.i.d$ samples uniformly drawn from $[n]$. In this case, the noise covariance is given by $\Sigma_0(\theta) = \Sigma_1(\theta) - \Sigma_2(\theta)$ with

$$\begin{aligned} \Sigma_1(\theta) &= \frac{1}{n} \sum_{i=1}^n \nabla \ell_i(\theta) \nabla \ell_i(\theta)^\top, \\ \Sigma_2(\theta) &= \nabla \mathcal{L}(\theta) \nabla \mathcal{L}(\theta)^\top. \end{aligned} \tag{7}$$

Since the relationship between $\Sigma_2(\theta)$ with $\nabla \mathcal{L}(\theta)$ is evident, the remaining task is to characterize the geometry of $\Sigma_1(\theta)$. Therefore, in the subsequent analysis, we shall refer $\Sigma(\theta)$ as $\Sigma_1(\theta)$ for simplicity.

Noting $\nabla \ell_i(\theta) = (f(x_i; \theta) - y_i) \nabla f(x_i; \theta)$, it follows that $\Sigma(\theta) = \frac{1}{n} \sum_{i=1}^n (f(x_i; \theta) - y_i)^2 \nabla f(x_i; \theta) \nabla f(x_i; \theta)^\top$. Then, the heuristic Hessian-based approximation (4) follows by assuming fitting errors $\{f(x_i; \theta) - y_i\}_{i=1}^n$ and model gradients $\{\nabla f(x_i; \theta)\}_{i=1}^n$ are decoupled (Mori et al., 2022).

**Over-parameterized linear models (OLMs).** An OLM is defined as $f(x; \theta) = F(\theta)^\top x$, where $F : \mathbb{R}^p \to \mathbb{R}^d$ denotes a general re-parameterization map. Although $f(\cdot; \theta)$ only represents linear functions, the associated loss landscape can be highly non-convex. Some typical examples include (i) linear model $F(w) = w$; (ii) diagonal linear network: $F(\theta) = (\alpha_1^2 - \beta_1^2, \ldots, \alpha_d^2 - \beta_d^2)^\top$; and (iii) linear network: $F(\theta) = W_1 W_2 \cdots W_L$. Notably, OLMs have been widely used to analyze the optimization and implicit bias of SGD (Arora et al., 2019; Woodworth et al., 2020; Pesme et al., 2021; HaoChen et al., 2021; Azulay et al., 2021; Li et al., 2022).

## 3 Loss alignment

By Eq. (2), to assess how the noise geometry affects the loss dynamics, we only need to estimate $\gamma(\theta) = \text{tr}(\Sigma_1(\theta) H(\theta)) - \nabla \mathcal{L}(\theta)^\top H(\theta) \nabla \mathcal{L}(\theta)$. It is evident that the second term is clear and what remains is to estimate $\gamma_1(\theta) = \text{tr}(\Sigma_1(\theta) G(\theta))$ by assuming the closeness between $G(\theta)$ and $H(\theta)$. If the approximation (4) holds, we have $\gamma_1(\theta) \approx 2\mathcal{L}(\theta) \|G(\theta)\|_F^2 =: \bar{\gamma}_1(\theta)$. We thus can define the following quantity to measure the influence of noise geometry on loss dynamics:

**Definition 3.1** (Loss alignment). $\mu(\theta) = \frac{\gamma_1(\theta)}{\bar{\gamma}_1(\theta)}$.

At global minima $\theta^*$, $\gamma_1(\theta) = \bar{\gamma}_1(\theta^*) = 0$ and we define $\gamma(\theta^*) = \frac{0}{0} = 1$ for convention. Under this definition, $\mu(\cdot)$ may not be continuous but we will show it holds that $\mu(\theta) \sim 1$.

### 3.1 (Over-parameterized) linear models

We first consider OLMs, for which we have:

**Theorem 3.2** (OLM)**.** *Consider OLMs and suppose Assumption 2.1 holds. For any $\epsilon, \delta \in (0, 1)$, if $n \gtrsim \max\{(d^2 \log^2 (1/\epsilon) + \log^2(1/\delta))/\epsilon, (d \log (1/\epsilon) + \log(1/\delta))/\epsilon^2\}$, then w.p. at least $1 - \delta$, it holds for any $\theta \in \mathbb{R}^p$ that $\frac{1-\epsilon}{(1+\epsilon)^2} \leq \mu(\theta) \leq \frac{3+\epsilon}{(1-\epsilon)^2}$.*

This theorem shows that the alignment strength is well-controlled and notably, the condition is *independent* from the re-parameterization map and the number of parameters. Hence, it can be effectively applied to linear networks regardless of the width and depth.

The following theorem shows that the sample size can be further relaxed for linear models.

**Theorem 3.3** (Linear model)**.** *Consider linear models and suppose Assumption 2.1 holds. For any $\epsilon, \delta \in (0, 1)$, if $n/\log(n/\delta) \gtrsim 1/\epsilon^2$ and $d_{\text{eff}} \gtrsim \log(n/\delta)/\epsilon^2$, then w.p. at least $1 - \delta$, it holds for any $\theta \in \mathbb{R}^d$ that $\frac{(1-\epsilon)^2}{(1+\epsilon)^2} \leq \mu(\theta) \leq \frac{(1+\epsilon)^2}{(1-\epsilon)^2}$.*

This theorem is established by leveraging the high dimensionality of inputs, as stated by the condition $d_{\text{eff}} \gtrsim \log n$. Notably, this condition includes the important regimes like $n \sim \log(d_{\text{eff}})$ (for sparse recovery) and $n \sim d_{\text{eff}}$ (the proportional scaling).

**Remark 3.4.** We remark that the two theorems are complementary for linear models. Specifically, consider the isotropic case where $S = I_d$. The two theorems together can cover $n \gtrsim 1$ (Theorem 3.3 holds for $n \lesssim e^d$, and Theorem 3.2 holds for $n \gtrsim d^2$).

### 3.2 Two-layer neural networks

Consider two-layer neural networks given by $f(x; \theta) = \sum_{k=1}^{m} a_k \phi(b_k^\top x)$ with $a_k \in \{\pm 1\}$ to be fixed. We use $\theta = (b_1^\top, \cdots, b_m^\top)^\top \in \mathbb{R}^{md}$ to denote the concatenation of all trainable parameters. Here, $\phi : \mathbb{R} \mapsto \mathbb{R}$ is an activation function with a non-degenerate derivative as defined below.

**Assumption 3.5.** There exist constants $\beta > \alpha > 0$ such that $\alpha \leq \phi'(z) \leq \beta$ holds for any $z \in \mathbb{R}$.

A typical nonlinear activation function that satisfies Assumption 3.5 is $\alpha$-Leaky ReLU: $\phi(z) = \max\{\alpha z, z\}$ with $0 < \alpha < 1$.

**Theorem 3.6** (Two-layer network)**.** *Consider the two-layer network $f(\cdot; \theta)$. Suppose Assumption 3.5 and Assumption 2.1 hold. For any $\epsilon, \delta \in (0, 1)$, if $n/\log(n/\delta) \gtrsim 1/\epsilon^2$ and $d_{\text{eff}} \gtrsim \log(n/\delta)/\epsilon^2$, then w.p. at least $1 - \delta$, it holds that for any $\theta \in \mathbb{R}^{md}$ that $\frac{\alpha^2(1-\epsilon)^2}{\beta^2(1+\epsilon)^2} \leq \mu(\theta) \leq \frac{\beta^2(1+\epsilon)^2}{\alpha^2(1-\epsilon)^2}$.*

This theorem establishes a uniform control for the alignment strength $\mu(\theta)$. Particularly, the number of samples required is *independent* of the network width $m$. The proof follows a similar approach to that of Theorem 3.3 and can be found in Appendix B. We remark that the conditions such as the non-degeneracy in activation function's derivatives are obligatory for establishing alignment across the *entire loss landscape*. In practice, such stringent conditions may not be necessary, as the focus is on regions navigated by SGD.

### 3.3 Numerical validations

Here we present small-scale experiments to validate our theoretical results with a 4-layer linear network and two-layer ReLU network (both layers are trainable). Both isotropic and anisotropic input distributions are examined and we set $n = 5 \log(d_{\text{eff}})$ to focus on the low-sample regime. The results are reported in Figure 1 and it is evident that across all examined scenarios, the alignment strength is consistently well-controlled and independent of the model size.

## 4 Directional alignment

In this section, we delve into a type of directional alignment: *whether noise energy along a given direction is proportional to the curvature of loss landscape along that direction*. Specifically, we use the following metric to quantify the alignment strength.

**Definition 4.1** (Directional alignment)**.** Given $v \in \mathbb{R}^p$, the alignment along $v$ is defined as $g(\theta; v) := \frac{v^\top \Sigma_1(\theta) v}{2\mathcal{L}(\theta)(v^\top G(\theta) v)}$, where $v^\top G(\theta) v$ denotes the curvature of local landscape along $v$.

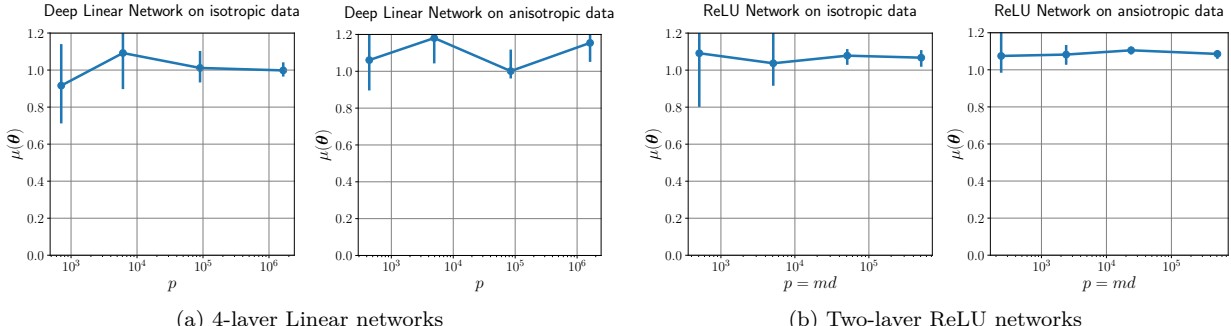

(a) 4-layer Linear networks          (b) Two-layer ReLU networks

Figure 1: The alignment strength $\mu(\theta)$ is close to 1 for various models across different model sizes. For all experiments, we set $n = 5 \log(d_{\text{eff}}), d_{\text{eff}} = 50$. The input data are drawn from $\mathcal{N}(0, S)$. For isotropic data, $S = I_{50}$; for anisotropic data, $S = \text{diag}(\lambda_1, \ldots, \lambda_D)$ with $\lambda_k = 1/\sqrt{k}$ for $k \in [D]$ where $D$ is chosen such that $d_{\text{eff}} = 50$. The error bar corresponds to the standard deviation over 20 independent runs. The targets are generated by a linear model $y_i = \langle w^*, x_i \rangle$, where $w^* \sim N(0, I_d)$. We compute $\mu(\theta)$ for randomly chosen $\theta$'s.

At global minima $\theta^*$, $\mathcal{L}(\theta^*) = 0, \Sigma_1(\theta^*) = 0$, and we define $g(\theta^*; v) = \frac{0}{0} = 1$ for convention.

Denoting by $z_t = \theta_t^\top v$ the component along $v$, we have

$$\mathbb{E}[z_{t+1}^2] = \mathbb{E}[\mathcal{G}_v(z_t)] + \eta^2 \mathbb{E}[(v^\top \xi_t)^2],$$

where $\mathcal{G}_v(z) = (z - \eta v^\top \nabla \mathcal{L}(\theta))^2$ represents the contribution from GD part. The additional term $\mathbb{E}[(v^\top \xi_t)^2]$ arises due to the injected noise, satisfying $\mathbb{E}[|\xi(\theta)^\top v|^2] = v^\top \Sigma_1(\theta_t)v - (v^\top \nabla \mathcal{L}(\theta_t))^2 = 2g(\theta_t, v)\mathcal{L}(\theta_t)v^\top G(\theta_t)v - (v^\top \nabla \mathcal{L}(\theta_t))^2$ by Definition 4.1. Therefore, the alignment defined above is crucial in describing the dynamics of SGD in subspaces.

**Theorem 4.2** (OLM). *Consider OLMs and suppose Assumption 2.1 holds. For any $\epsilon, \delta \in (0, 1)$, if $n \gtrsim \max\{(d^2 \log^2(1/\epsilon) + \log^2(1/\delta))/\epsilon, (d \log(1/\epsilon) + \log(1/\delta))/\epsilon^2\}$, then w.p. at least $1 - \delta$, it holds for any $\theta, v \in \mathbb{R}^p$ that $\frac{1-\epsilon}{(1+\epsilon)^2} \leq g(\theta; v) \leq \frac{3+\epsilon}{(1-\epsilon)^2}$.*

This theorem establishes a uniform control for the alignment across all directions and the entire parameter space. Notably, the condition is independent of the degree of overparameterization. The subsequent theorem shows that if focusing on some specific directions, the sample size $n$ can be further relaxed.

**Theorem 4.3** (Linear model). *Consider linear models and suppose Assumption 2.1 holds. Let $\mathcal{V} = \{v_1, \cdots, v_K\}$ be $K$ fixed directions of interest. For any $\epsilon \in (0, 1/2)$, there exist $C_1 = C_1(\epsilon) > 0$ and $C_2 = C_2(\epsilon) > 0$ such that if $n \gtrsim C_1 d \log d$, then w.p. at least $1 - \frac{KC_2}{n^2} - \epsilon^d$, it holds for any $\theta \in \mathbb{R}^p$ and $v \in \mathcal{V}$ that $\frac{1-\epsilon}{(1+\epsilon)^2} \leq g(\theta; v) \leq \frac{3+\epsilon}{(1-\epsilon)^2}$.*

It is worth noting that the above theorems establish the directional alignment for the *entire parameter space*. However, in practice, what matter are typical regions navigated by SGD. This is the gap between our theory and the practice.

**Numerical validations.** In this experiment, we consider the alignment along the eigen-directions of Hessian matrix. Let $G(\theta) = \sum_k \lambda_k u_k u_k^\top$ be the eigen-decomposition of $G(\theta)$ where $\{\lambda_k\}_k$ are the eigenvalues in a decreasing order and $\{u_k\}$ are the corresponding eigen-directions. Here we omit the dependence of $\theta$ in $\{(\lambda_k, u_k)\}_k$ for brevity. Note that $\lambda_k$ is the curvature of local landscape along $u_k$. Denote by $\alpha_k = \mathbb{E}[|\xi^\top u_k|^2]/(2\mathcal{L}(\theta))$ the relatively size of noise energy along $u_k$. Then, the directional alignment (4.1) satisfies $g(\theta, u_k) = \alpha_k/\lambda_k$.

In Figure 2a, we examine with linear regression under the regimes of limited data, which is beyond our theorems. We still observed that $g(\theta, v)$ is close 1 for all eigen directions $v$, which is consistent with our theoretical findings. In Figure 2b, we further consider the classification of CIFAR-10 with a small convolutional neural network (CNN) and fully-connected neural network (FNN). We can see that the observation is consistent with

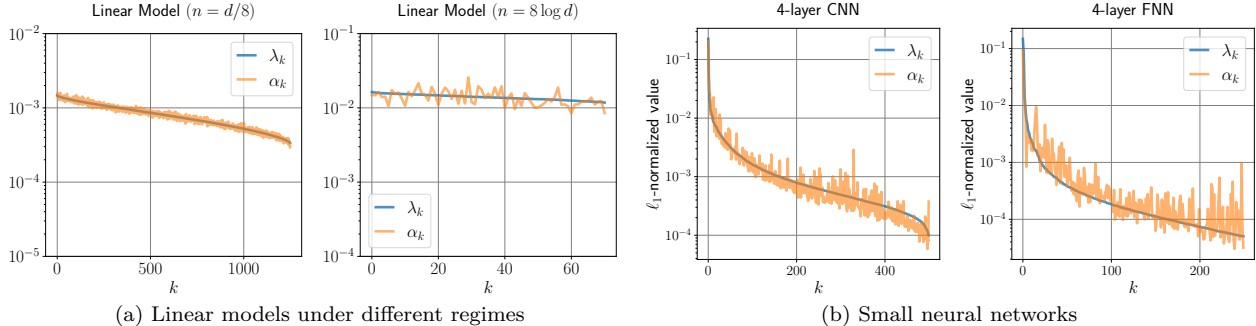

(a) Linear models under different regimes  (b) Small neural networks

Figure 2: How the components of noise energy in *eigen-directions* $\{\alpha_k\}_k$ are proportional to the corresponding curvatures $\{\lambda_k\}_k$. $\alpha_k/\lambda_k$ can reflect the directional alignment (4.1) along the eigen-directions. (a) Linear models on Gaussian data in the regimes with limited data, where we fix $d = 10^4$ and set $n$ accordingly ($n = d/8, n = 8\log d$). (b) 4-layer CNN and 4-layer FNN on CIFAR-10 dataset. For more experimental details, we refer to Appendix A.

Figure 2a, where the alignments in all eigen-directions are well-controlled, though the fluctuations become more significant.

## 5 Applications

To illustrate the utility of our characterization of noise geometry, this section presents a fine-grained analysis of both the convergence and escape behaviors of SGD by leveraging the loss alignment and directional alignment properties of SGD noise.

### 5.1 Convergence rate of SGD

In this section, leveraging the loss alignment property of SGD noise, we provide a refined convergence analysis of SGD.

For clarity, we focus on the case of *high-dimensional linear regression*, while the generalization to non-convex problems under typical smoothness and the Polyak-Łojasiewicz (PL) condition is straightforward. In strongly convex linear regression, we have $f(x; \theta) = \theta^\top x$, where $H(\theta) = G(\theta) \equiv \frac{1}{n}\sum_{i=1}^n x_i x_i^\top$ with $\lambda_{\min}(G) > 0$.

Recalling Remark 3.4, the loss alignment of SGD noise holds with high probability for linear regression, as supported by both theoretical and empirical results in Section 3. For the sake of clarity, we explicitly state it here as an assumption.

**Assumption 5.1.** It holds that for any $\theta \in \mathbb{R}^d$, $\mu(\theta) \leq 3.1$, where the loss alignment factor $\mu(\theta)$ is defined in Definition 3.1.

**Theorem 5.2** (Convergence of SGD). *Suppose Assumption 5.1 holds and use the constant learning rate* $\eta^* = \lambda_{\min}(G)/(2\|G\|_F^2)$. *Then SGD converges with* $\mathbb{E}[\mathcal{L}(\theta_t)] \leq \left(1 - \frac{\lambda_{\min}^2(G)}{2\|G\|_F^2}\right)^t \mathbb{E}[\mathcal{L}(\theta_0)]$.

|  | convergence rate |
|---|---|
| traditional result | $\mathcal{O}\left(1/t\right)$ |
| Bassily et al. (2018) | $\left(1 - \frac{\lambda_{\min}^2(G)}{2\|G\|_2(\max_{i\in[n]}\|x_i\|^2)}\right)^t$ |
| this work | $\left(1 - \frac{\lambda_{\min}^2(G)}{2\|G\|_F^2}\right)^t$ |

Table 1: Comparison of the convergece rates of SGD for strongly convex linear regression.

Theorem 5.2 demonstrates that SGD converges exponentially fast, with a coefficient of $\lambda_{\min}^2(G)/\|G\|_F^2$. The proof can be found in Appendix D. Notably, this result contrasts sharply with previous findings, as shown in Table 1.

- In classical convergence analysis of SGD, the bounded variance assumption $\mathbb{E}[\|\xi_t\|^2] \le \sigma^2$ is typically imposed. Under this setting, even for strongly convex functions, SGD must use a decay learning rate to ensure convergence, resulting in a slow polynomial rate $\mathcal{O}(1/t)$, even the problem is strongly convex (Hazan et al., 2016).

- Recently, Bassily et al. (2018) showed that in the interpolation regime, SGD noise diminish to zero at global minima, and satisfy $\mathbb{E}_\xi[\|\nabla \ell_\xi(\theta)\|^2] \le \sigma^2 \mathcal{L}(\theta)$, where $\sigma^2$ denotes the maximal smoothness of individual loss $\ell_i$. Leveraging this property, they showed that SGD can converge with a constant learning rate and specifically, for linear regression, it is shown that $\sigma^2 = \max_{i \in [n]} \|x_i\|^2$, and the exponential coefficient is $\lambda_{\min}^2(G)/(2\|G\|_2 (\max_{i \in [n]} \|x_i\|^2))$.

- However, as explained above, the analysis in Bassily et al. (2018) exploits solely the particular structure of noise magnitude, overlooking the shape structure. Consequently, Bassily et al. (2018) underestimates the largest learning rate that can be used. In contrast, our noise characterization (Assumption 5.1) allows us to consider both structures in a unified manner, enabling a more accurate estimate of the convergence rate, characterized by the coefficient $\lambda_{\min}^2(G)/\|G\|_F^2$. Note that

$$\|G\|_2 \left(\max_{i \in [n]} \|x_i\|^2\right) \ge \|G\|_2 \left(\frac{1}{n} \sum_{i \in [n]} \|x_i\|^2\right)$$
$$= \|G\|_2 \operatorname{Tr}(G) \ge \|G\|_F^2.$$

Thus, our convergence rate is much faster than that of Bassily et al. (2018) in two distinct scenarios: (i) $\max_{i \in [n]} \|x_i\|^2 \gg \frac{1}{n} \sum_{i \in [n]} \|x_i\|^2$, which occurs with high probability for sufficient samples (large $n$); (ii) $\|G\|_2 \operatorname{Tr}(G) \gg \|G\|_F^2$. For example, if the eigenvalues of $G$ decay as $\lambda_k = k^{-\alpha}$ for $1/2 < \alpha < 1$ in high-dimensional settings, then $\|G\|_2 \operatorname{Tr}(G) = \sum_{k=1}^d \lambda_k \to \infty$ as $d \to \infty$, while $\|G\|_F^2 = \sum_{k=1}^d \lambda_k^2$ remains bounded as $d \to \infty$.

## 5.2 How SGD escapes from sharp minima

In this section, we provide a fine-grained analysis of how SGD escapes from sharp minima by leveraging the directional alignment of noise geometry. In contrast to existing analyses (Zhu et al., 2019; Mori et al., 2022; Wu et al., 2022; Xie et al., 2020; Kleinberg et al., 2018) which only considered the escape rate, we also delve into the escape directions.

Let $\theta^*$ be a minimum of interest and consider the interpolation regime, i.e., $\mathcal{L}(\theta^*) = 0$. The local escape behavior can be characterized by linearizing the SGD dynamics, which corresponds to the linearized model $f(\cdot; \theta) \approx f(\cdot; \theta^*) + \langle \nabla f(\cdot; \theta^*), \theta - \theta^* \rangle$. We refer to (Wu et al., 2022, Section 3.2) for more details. Thus, without loss of generality, we can simply consider the linearized model in the subsequent analysis. Let $w = \theta - \theta^*$ and $G(\theta^*) = \frac{1}{n} \sum_{i=1}^n \nabla f(x_i; \theta^*) \nabla f(x_i; \theta^*)^\top$. Then, for the linearized model, we have $\mathcal{L}(w) = \frac{1}{2} w^\top G(\theta^*) w$ and $\nabla \mathcal{L}(w) = G(\theta^*) w$. Thus, the linearized SGD updates as follows

$$w_{t+1} = w_t - \eta \left(G(\theta^*) w_t + \xi_t\right),$$

where $\xi_t$ is the SGD noise. We make the following assumption on the noise geometry.

**Assumption 5.3** (Eigen-directional alignment). let $G(\theta^*) = \sum_{i=1}^p \lambda_i u_i u_i^\top$ be the eigen decomposition of $G(\theta^*)$. Assume that there exist $A_1, A_2 > 0$ such that it holds for any $w \in \mathbb{R}^p$

$$A_1 \mathcal{L}(w) \lambda_i \le \mathbb{E}[|\xi(w)^\top u_i|^2] \le A_2 \mathcal{L}(w) \lambda_i.$$

Section 4 has provided both theoretical and experimental evidence in support of this assumption. For the sake of clarity, we explicitly state it here as an underlying assumption.

**Eigen-decomposition of SGD.** By leveraging Assumption 5.3, we can analyze the SGD dynamics in the eigenspace. Let $w_t = \sum_{i=1}^d w_{t,i} u_i$ with $w_{t,i} = u_i^\top w_t$. Then, $w_{t+1,i} = (1 - \eta \lambda_i) w_{t,i} + \eta \xi_t^\top u_i$. Taking the expectation of the square of both sides, we obtain

$$\mathbb{E}[w_{t+1,i}^2] = (1 - \eta \lambda_i)^2 \mathbb{E}[w_{t,i}^2] + \eta^2 \mathbb{E}[|u_i^\top \xi_t|^2], \tag{8}$$

where the noise term: $\mathbb{E}[|u_i^\top \xi_t|^2] \sim \lambda_i \mathcal{L}(w_t)$ according to Assumption 5.3.

Let $X_t = \sum_{i=1}^k \lambda_i \mathbb{E}[w_{t,i}^2], Y_t = \sum_{i=k+1}^d \lambda_i \mathbb{E}[w_{t,i}^2]$, denoting the components of loss energy along sharp and flat directions, respectively. Let $D_{t,k} = Y_t/X_t$, which measures the concentration of loss energy along flat directions. Analogously, let $P_{t,k} = \sum_{i=k+1}^d \mathbb{E}[w_{t,i}^2]/\sum_{i=1}^k \mathbb{E}[w_{t,i}^2]$, which measures the concentration of variance along flat directions. It is easy to show that $P_{t,k} \geq D_{t,k}\lambda_k/\lambda_{k+1}$. Therefore, when $\lambda_k/\lambda_{k+1}$ is lower bounded, a concentration of loss energy along flat directions can lead to a similar concentration in terms of variance.

**Theorem 5.4** (Escape of SGD). *Suppose Assumption 5.3 holds and let $\eta = \frac{\beta}{\|G(\theta^*)\|_F}$. Then, there exists absolute constants $c_1, c_2 > 0$ such that if $\beta \geq c_1$, then SGD will escape from the minimum $\theta^*$: $\lim_{t\to\infty} \|\theta_t - \theta^*\| = \infty$. Moreover, for any $k \in [d]$, it holds that when $t \geq \max\left\{1, \frac{\log\left(c_2/\eta(\sum_{i=1}^k \lambda_i^2)^{1/2}\right)}{\log \beta}\right\}$: $D_{t,k} \gtrsim \frac{\sum_{i=k+1}^d \lambda_i^2}{\sum_{i=1}^k \lambda_i^2}$.*

The proof can be found in Appendix E. This theorem reveals that during SGD's escape process, the loss rapidly accumulates a significant component along flat directions of the loss landscape. The precise loss ratio between the flat and sharp directions is governed by the spectrum of Hessian matrix. In particular, $D_{t,1} \gtrsim \mathrm{srk}(G^2) - 1$, indicating that in high dimension, i.e., $\mathrm{srk}(G^2) \gg 1$, the loss energy along the sharpest directions becomes negligible during the SGD's escape process. This stands in stark contrast to GD, which always escapes along the sharpest direction:

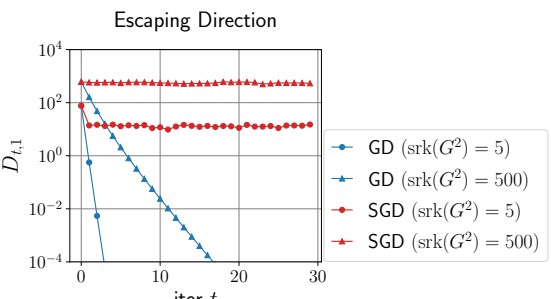

Figure 3: Comparison of escape directions between SGD and GD. The problem is linear regression and both SGD and GD are initialized near the global minimum by $w_0 \sim \mathcal{N}(w^*, e^{-10}I_d/d)$. To ensure escape, we choose $\eta = 1.2/\|G\|_F$ and $\eta = 4/(\lambda_1 + \lambda_2)$ for SGD and GD, respectively. Please refer to Appendix A for more experimental details.

**Proposition 5.5** (Escape of GD). *Consider GD with learning rate $\eta = \beta/\lambda_1$. If $\beta > 2$, then $D_{t,1} \leq \sum_{i=2}^d \frac{\lambda_i(1-\eta\lambda_i)^{2t}w_{0,i}^2}{\lambda_1(1-\eta\lambda_1)^{2t}w_{0,1}^2}$.*

In particular, if $w_{0,1} \neq 0$ and $\lambda_1 > \lambda_2$, then the above proposition implies that $D_{t,1}$ decreases to 0 exponentially fast for GD.

Figure 3 presents numerical comparisons of the escaping directions between SGD and GD. It is evident that $D_{t,1}$ exponentially decreases to zero for GD, indicating that GD escapes along the sharpest direction. In contrast, for SGD, $D_{t,1}$ remains significantly large, indicating that SGD retains a substantial component along the flat directions during the escape process. Furthermore, the value of $D_{t,1}$ positively correlates with $\mathrm{srk}(G^2)$, as predicted by our Theorem 5.4. These observations provide empirical confirmation of our theoretical findings.

**Explaining the implicit bias of cyclical learning rate.** Gaining insights into the escape direction can be valuable for understanding how SGD explores the non-convex landscape. Specifically, we provide a concrete example to illustrate the role of escape direction in enhancing the implicit bias of SGD through Cyclical Learning Rate (CLR) (Smith, 2017; Loshchilov & Hutter, 2017). As shown in Figure 2 of Huang et al. (2018), utilizing CLR enables SGD to cyclically escapes from (when increasing LR) and slides into (when decreasing LR) sharp regions, ultimately progressing towards flatter minima. We hypothesize that escape along flat directions plays a pivotal role in guiding SGD towards flatter region in this process.

Following Ma et al. (2022), we consider a toy OLM $f(x; w) = (w_2/\sqrt{w_1^2 + 1})x$ with $x \sim \mathcal{N}(0, 1)$. For simplicity, we consider the online setting, where the landscape is

$$\mathcal{L}(w) = w_2^2/[2(w_1^2 + 1)].$$

The global minima valley is $S = \{w : w_2 = 0\}$ and for $w \in S$, $\mathrm{tr}[\nabla^2 \mathcal{L}(w)] = 1/(1 + w_1^2)$. Hence, the minimum gets flatter along the valley $S$ when $|w_1|$ grows up. In Figure 4, we visualize the trajectories for both SGD+CLR and GD+CLR. One can observe that

- SGD escape from the minima along both the flat direction $e_1$ and sharp direction $e_2$. The component of along $e_1$ leads to considerable increase in $w_1^2$, facilitating the movement towards flatter region along the minimum valley $S$.

- On the contrary, GD escapes only along $e_2$, yielding no increase in $w_1^2$. Thus, we cannot observe clear movement towards flatter region for GD+CLR.

Thus, in this toy model, the fact that SGD escapes along flat directions is crucial in amplifying the implicit bias towards flat minima.

Nonetheless, understanding how the above mechanism manifests in practice remains an open question that warrants further investigation. We defer this topic for future work, as the primary focus of this paper is to understand the noise geometry rather than exhaustively explore its applications.

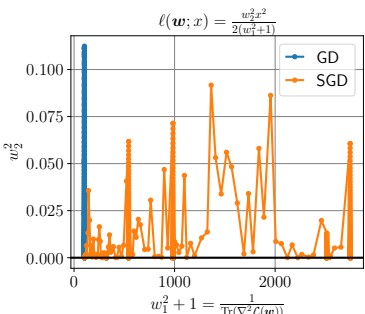

Figure 4: Visualization of the trajectories of SGD+CLR v.s. GD+CLR for our toy model. Both cases use the same CLR schedule. We can observe that SGD+CLR moves significantly towards flatter region, while GD+CLR only oscillates along the sharpest direction. We have extensively tuned the learning rates for GD+CLR but do not observe significant movement towards flatter region in any case.

## 6 Larger-scale experiments for deep neural networks

We have already provided small-scale experiments to confirm our theoretical findings. We now turn to justify the practical relevance of theoretical findings by examining the classification of CIFAR-10 dataset (Krizhevsky & Hinton, 2009) with practical VGG nets (Simonyan & Zisserman, 2015) and ResNets (He et al., 2016). Note that larger-scale experiments on loss alignment have been previously presented in Wu et al. (2022). Thus, our focus here is on investigating the directional alignment of SGD. We refer to Appendix A for experimental details.

**The directional alignment along eigen-directions.** Figure 5 presents the directional alignments of SGD noise for ResNet-38 and VGG-13. The alignment is examined along the eigen-directions of the local landscape. The two quantities: $\lambda_k$ and $\alpha_k$ under $\ell_1$ normalization (i.e., $\lambda_k / \|\lambda\|_1$ and $\alpha_k / \|\alpha\|_1$) are plotted. Here, $\lambda_k$ and $\alpha_k$ represent the curvature and the component of noise energy along the $k$-th eigen-direction, respectively. One can see that the alignment between $\alpha_k$ and $\lambda_k$ still exists for ResNet-38 and VGG-13, but the ratio between them becomes significantly larger. As a comparison, we refer to Figure 2b, where the ratio is well-controlled for small-scale networks trained for classifying the same dataset. We hypothesize that this observation is consistent with our theoretical results in Section 4: one-sided bounds require much less samples.

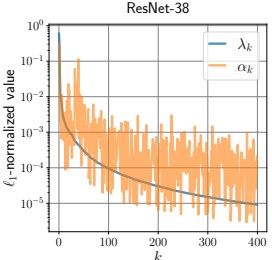 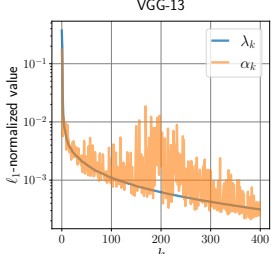

Figure 5: Three distributions ($\{\lambda_k\}_k$ and $\{\alpha_k\}_k$) for larger-scale neural networks, which reflect the directional alignment (4.1) along the eigen directions of the local landscape.

**The escape direction of SGD.** For large models, it is computationally prohibitive to compute the quantity $D_{t,k}$ since it needs to compute the whole spectrum. Thus, we consider to measure the component along different

directions without reweighting. Let $\theta^*$ be the minimum of interest and $\theta_t$ be SGD/GD solution at step $t$. Define $p_{t,k} = \langle \theta_t - \theta^*, u_1 \rangle$ for $k = 1$ and $p_{t,k} = (\sum_{i=1}^{k} \langle \theta_t - \theta^*, u_i \rangle^2)^{1/2}$ for $k > 1$; $r_{t,k} = (\|\theta_t - \theta^*\|^2 - p_{t,k}^2)^{1/2}$. Notably, $p_{t,k}$ and $r_{t,k}$ represent the component along sharp and flat directions, respectively.

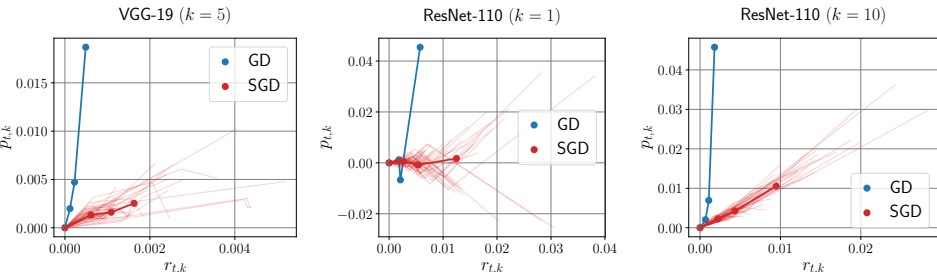

Figure 6: The red curves are 50 escaping trajectories of SGD and their average; the blue curves corresponding to GD. The sharp minimum $\theta^*$ is found by SGD. Then, we run SGD and GD starting from $\theta^*$ and the learning rates are tuned to ensure escaping.

In Figure 6, we plot $(p_{t,k}, r_{t,k})$ for VGG-19 and ResNet-110, where we examine various $k$ values. The plots clearly demonstrate that the escape direction of SGD exhibits significant components along the flat directions. On the other hand, GD tends to escape along much sharper directions. These empirical findings align well with our theoretical findings in Section 5.2.

## 7 Concluding remark

In this paper, we present a thorough investigation of the geometry of SGD noise, providing quantitative characterizations of how SGD noise aligns well with the landscape's local geometry. Furthermore, we explore the implications of these findings by analyzing both SGD convergence rate and the direction of SGD escaping from sharp minima, as well as its role in enhancing the implicit bias toward flatter minima through cyclical learning rate.

Generally speaking, understanding the noise geometry is crucial for comprehending many aspects of SGD dynamics. Our analysis of SGD's escape direction only serves as a preliminary example of this. Looking ahead, there are numerous potential avenues for further research. One such direction is to investigate how noise geometry influences the convergence and stability of the loss in SGD, as explored in recent works Thomas et al. (2020); Wu et al. (2022); Wu & Su (2023). Furthermore, as indicated by the analysis in Section 5.2, SGD noise can substantionally alter the unstable directions. This observation opens up the intriguing possibility of studying the impact of noise geometry on the Edge of Stability (EoS) (Cohen et al., 2020; Wu et al., 2018) and the associated unstable convergence phenomena of SGD (Ahn et al., 2022).

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

## A  Experimental Setups

In this section, we provide the experiment details for directional alignment experiments (in Figure 2 and Figure 5) and escaping experiments (in Figure 3 and Figure 6).

**Small-scale experiments** (Figure 2 and 3).

- In Figure 2 (a), we conduct directional alignment experiments on linear regression. The inputs $\{x_i\}_{i=1}^n$ are drawn from $\mathcal{N}(0, I_d)$. The targets are generated by a linear model, i.e., $y_i = w^{*\top} x_i$, where $w^* \sim N(0, I_d)$. We fix $d = 10^4$ and change $n$ accordingly ($n = d/8, n = 8 \log d$). Regarding the parameter $\theta$, it is drawn from $\mathcal{N}(0, I_p)$.

- In Figure 2 (b), we conduct directional alignment experiments on a 4-layer CNN ($p = 43,072$) and a 4-layer FNN ($p = 219,200$). Specifically, the architecture of 4-layer CNN is $\texttt{Conv}(3,6,5) \to \texttt{ReLU} \to \texttt{MPool}(2,2) \to \texttt{Conv}(6,16,5) \to \texttt{ReLU} \to \texttt{MPool}(2,2) \to \texttt{Linear}(400,100) \to \texttt{ReLU} \to \texttt{Linear}(100,2)$. and the 4-layer FNN is a ReLU-activated fully-connected network with the architecture: $784 \to 256 \to 64 \to 32 \to 2$. We use the CIFAR-10 dataset with label$=0, 1$. Regarding the parameter $\theta$, it is drawn from $\mathcal{N}(0, I_p)$.

- In Figure 3, we conduct escaping experiments on linear regression with $w^* = 0$. Both SGD and GD are initialized near the global minimum by $w_0 \sim \mathcal{N}(0, e^{-10} I_d/d)$. To ensure escaping, we choose $\eta = 1.2/\|G\|_{\mathrm{F}}$ and $\eta = 4/(\lambda_1 + \lambda_2)$ for SGD and GD, respectively. We fix $n = 10^5$ and $d = 10^3$, and the inputs $\{x_i\}_{i=1}^n$ are drawn from $\mathcal{N}(0, \mathrm{diag}(\lambda)/d)$, where $\lambda \in \mathbb{R}^d$ and $\lambda_1 \geq \lambda_2 = \cdots = \lambda_d \geq 0$. Moreover, we set $\lambda_1 = 1$ change $\lambda_2$ accordingly to obtain different $\mathrm{srk}(G^2)$.

**Larger-scale experiments** (Figure 5 and 6).

- Dataset. For the experiments in Figure 5 and 6, we use the CIFAR-10 dataset with label=0, 1 and the full CIFAR-10 dataset to train our models, respectively.

- Models. We conduct experiments on large-scale models: ResNet-38 ($p = 558, 222$), VGG-13 ($p = 605, 458$), ResNet-110 ($p = 1, 720, 138$), and VGG-19 ($p = 20, 091, 338$). Specifically, we use standard ResNets (He et al., 2016) and VGG nets (Simonyan & Zisserman, 2015) without batch normalization. For ResNets, we follow Zhang et al. (2019) to use the fixup initialization in order to ensure that the model can be trained without batch normalization.

- Training. All explicit regularizations (including weight decay, dropout, data augmentation, batch normalization, learning rate decay) are removed, and a simple constant-LR SGD is used to train our models. Specifically, all these models are trained by SGD with learning rate $\eta = 0.1$ and batch size $B = 32$ until the training loss becomes smaller than $10^{-4}$.

**Efficient computations** of the top-$k$ eigen-decomposition of $G$ and $\Sigma$. We utilize the functions `eigsh` and `LinearOperator` in `scipy.sparse.linalg` to calculate top-$k$ eigenvalues and eigenvectors of $G$ and $\Sigma$, and the key step is to efficiently calculate $Gv$ and $\Sigma v$ for any given $v \in \mathbb{R}^p$.

- For small-scale experiments, they can be calculated directly.

- For the large-scale models, we need further approximations since the computation complexity $\mathcal{O}(np)$ is prohibitive in this case. To illustrate our method, we will use $Gv$ as an example and apply a similar approach to $\Sigma v$. Notice that the formulation $Gv = \frac{1}{n} \sum_{i=1}^{n} (x_i^\top v) x_i$ are all in the form of sample average, which allows us to perform Monte-Carlo approximation. Specifically, we randomly choose $b$ samples $\{x_{i_j}\}_{j=1}^{b}$ from $x_1, \ldots, x_n$ and use $\frac{1}{b} \sum_{j=1}^{b} (x_{i_j}^\top v) x_{i_j}$ estimate $Gv$, with the computation complexity $\mathcal{O}(bp)$. For the experiments on CIFAR-10, we test $b$'s with different values and find that $b = 2k$ is sufficient to obtain a reliable approximation of the top-$k$ eigenvalues and eigenvectors. Hence, for all large-scale experiments in this paper, we use $b = 2k$ to speed up the computation of the top-$k$ eigenvalues and eigenvectors.

Larger-scale experiments are conducted on 1 3080 GPU, while small-scale experiments are conducted on CPU.

## B  Proofs of Section 3

### B.1  Proof of Theorem 3.2

This result is a direct corollary of Theorem 4.2, which is proved in Appendix C.

By Theorem 4.2, under the same conditions, the following uniform bound holds:

$$\frac{1 - \epsilon}{(1 + \epsilon)^2} \leq \inf_{\theta, v \in \mathbb{R}^p} g(\theta; v) \leq \sup_{\theta, v \in \mathbb{R}^p} g(\theta; v) \leq \frac{3 + \epsilon}{(1 - \epsilon)^2},$$

which means that for any $\theta \in \mathbb{R}^p, v \in \mathbb{S}^{p-1}$, we have

$$\frac{1 - \epsilon}{(1 + \epsilon)^2} \cdot 2\mathcal{L}(\theta) v^\top G(\theta) v \leq v^\top \Sigma_1(\theta) v \leq \frac{3 + \epsilon}{(1 - \epsilon)^2} \cdot 2\mathcal{L}(\theta) v^\top G(\theta) v.$$

Consider the orthogonal decomposition of $G(\theta)$: $G(\theta) = \sum_{k=1}^{p} \lambda_k u_k u_k^\top$. Notice that

$$\mathrm{Tr}(\Sigma_1(\theta) G(\theta)) = \sum_{k=1}^{p} \lambda_k u_k^\top \Sigma_1(\theta) u_k,$$

$$\|G(\theta)\|_{\mathrm{F}}^2 = \mathrm{Tr}(G(\theta) G(\theta)) = \sum_{k=1}^{p} \lambda_k u_k^\top G(\theta) u_k.$$

Then we obtain

$$\mathrm{Tr}(\Sigma_1(\theta)G(\theta)) \geq \frac{1-\epsilon}{(1+\epsilon)^2} \cdot 2\mathcal{L}(\theta) \sum_{k=1}^{p} \lambda_k u_k^\top G(\theta)u_k = \frac{1-\epsilon}{(1+\epsilon)^2} \cdot 2\mathcal{L}(\theta) \|G(\theta)\|_{\mathrm{F}}^2,$$

$$\mathrm{Tr}(\Sigma_1(\theta)G(\theta)) \leq \frac{3+\epsilon}{(1-\epsilon)^2} \cdot 2\mathcal{L}(\theta) \sum_{k=1}^{p} \lambda_k u_k^\top G(\theta)u_k = \frac{3+\epsilon}{(1-\epsilon)^2} \cdot 2\mathcal{L}(\theta) \|G(\theta)\|_{\mathrm{F}}^2,$$

which means $\frac{1-\epsilon}{(1+\epsilon)^2} \leq \mu(\theta) \leq \frac{3+\epsilon}{(1-\epsilon)^2}$. From the arbitrariness of $\theta$, we complete the proof. $\qquad\square$

## B.2 Proof of Theorem 3.3

For the linear model, i.e., $\theta = w$ and $F(w) = w$ in OLMs, we have the following lower bound:

$$\begin{aligned}
\mu(w) &= \frac{\mathrm{Tr}\left(\Sigma_1(w)G(w)\right)}{2\mathcal{L}(w)\|G(w)\|_{\mathrm{F}}^2} \\
&= \frac{\mathrm{Tr}\left(\left(\frac{1}{n}\sum_{j=1}^{n} x_j x_j^\top\right)\left(\frac{1}{n}\sum_{i=1}^{n}(F(\theta)^\top x_i)^2(\nabla F(\theta)^\top x_i)(\nabla F(\theta)^\top x_i)^\top\right)\right)}{\left(\frac{1}{n}\sum_{i=1}^{n}(F(\theta)^\top x_i)^2\right)\left(\frac{1}{n^2}\sum_{i=1}^{n}\sum_{j=1}^{n}\left(x_i^\top\nabla F(\theta)\nabla F(\theta)^\top x_j\right)^2\right)} \\
&= \frac{\frac{1}{n^2}\sum_{i=1}^{n}\sum_{j=1}^{n}(w^\top x_i)^2\left(x_i^\top x_j\right)^2}{\left(\frac{1}{n}\sum_{i=1}^{n}(w^\top x_i)^2\right)\left(\frac{1}{n^2}\sum_{i=1}^{n}\sum_{j=1}^{n}\left(x_i^\top x_j\right)^2\right)} \geq \frac{\left(\frac{1}{n}\sum_{i=1}^{n}(w^\top x_i)^2\right)\left(\min_{i\in[n]}\frac{1}{n}\sum_{j=1}^{n}\left(x_i^\top x_j\right)^2\right)}{\left(\frac{1}{n}\sum_{i=1}^{n}(w^\top x_i)^2\right)\left(\frac{1}{n^2}\sum_{i=1}^{n}\sum_{j=1}^{n}\left(x_i^\top x_j\right)^2\right)} \\
&= \frac{\min_{i\in[n]}\frac{1}{n}\sum_{j=1}^{n}\left(x_i^\top x_j\right)^2}{\max_{i\in[n]}\frac{1}{n}\sum_{j=1}^{n}\left(x_i^\top x_j\right)^2} \geq \frac{\min_{i\in[n]}\|x_i\|^4 + (n-1)\min_{i\in[n]}\frac{1}{n-1}\sum_{j\neq i}(x_i^\top x_j)^2}{\max_{i\in[n]}\|x_i\|^4 + (n-1)\max_{i\in[n]}\frac{1}{n-1}\sum_{j\neq i}(x_i^\top x_j)^2}.
\end{aligned} \tag{9}$$

In the same way, the upper bound holds:

$$\mu(w) = \frac{\mathrm{Tr}\left(\Sigma_1(w)G(w)\right)}{2\mathcal{L}(w)\|G(w)\|_{\mathrm{F}}^2} \leq \frac{\max_{i\in[n]}\frac{1}{n}\sum_{j=1}^{n}\left(x_i^\top x_j\right)^2}{\min_{i\in[n]}\frac{1}{n}\sum_{j=1}^{n}\left(x_i^\top x_j\right)^2} \leq \frac{\max_{i\in[n]}\|x_i\|^4 + (n-1)\max_{i\in[n]}\frac{1}{n-1}\sum_{j\neq i}(x_i^\top x_j)^2}{\min_{i\in[n]}\|x_i\|^4 + (n-1)\min_{i\in[n]}\frac{1}{n-1}\sum_{j\neq i}(x_i^\top x_j)^2}.$$

Then we only need to estimate $\|x_i\|^4$ and $\frac{1}{n-1}\sum_{j\neq i}(x_i^\top x_j)^2$ for each $i \in [n]$, respectively.

**Step I: Estimation of $\|x_i\|^4$.** Let $z_i = S^{-1/2}x_i$, then $\|x_i\|^2 = z_i^\top S z_i$ and $z_1, \cdots, z_n \overset{\text{i.i.d.}}{\sim} \mathcal{N}(0, I_d)$.

For a fix $i \in [n]$, by Lemma F.2, there exists an absolute constant $C_1 > 0$ such that for any $\epsilon \in (0,1)$, we have

$$\mathbb{P}\left(\left|z_i^\top S z_i - \mathrm{Tr}(S)\right| \geq \epsilon\,\mathrm{Tr}(S)\right) \leq 2\exp\left(-C_1\min\left\{\frac{\epsilon^2\,\mathrm{Tr}^2(S)}{\|S\|_{\mathrm{F}}^2}, \frac{\epsilon\,\mathrm{Tr}(S)}{\|S\|_2}\right\}\right).$$

Noticing that $\mathrm{Tr}(S)\|S\|_2 = \lambda_1\sum_i \lambda_i \geq \sum_i \lambda_i^2 = \|S\|_F$, we thus have

$$\frac{\mathrm{Tr}^2(S)}{\|S\|_{\mathrm{F}}^2} \geq \frac{\mathrm{Tr}(S)}{\|S\|_2} = \mathrm{srk}(S).$$

Therefore,

$$\mathbb{P}\left(\left|z_i^\top S z_i - \text{Tr}(S)\right| \ge \epsilon \text{Tr}(S)\right) \le 2\exp\left(-C_1 \frac{\text{Tr}(S)}{\|S\|_2} \min\left\{\epsilon, \epsilon^2\right\}\right) = 2\exp\left(-C_1 \epsilon^2 \text{srk}(S)\right).$$

Applying a union bound over all $i \in [n]$, we have

$$\mathbb{P}\left(\left|\|x_i\|^2 - \text{Tr}(S)\right| \ge \epsilon \text{Tr}(S), \forall i \in [n]\right) \le 2n\exp\left(-C_1 \epsilon^2 \text{srk}(S)\right).$$

In the other word, for any $\epsilon, \delta \in (0,1)$, if $\text{srk}(S) \gtrsim \log(n)/\epsilon^2$, then $w.p.$ at least $1 - \delta/3$, we have

$$(1-\epsilon)^2 \le \frac{\|x_i\|_2^4}{\text{Tr}^2(S)} \le (1+\epsilon)^2, \ \forall i \in [n].$$

**Step II: Estimation of $\frac{1}{n-1}\sum\limits_{j \ne i}(x_i^\top x_j)^2$.** First, we fix $i \in [n]$. Notice that $(x_i^\top x_j)^2$ $(j \ne i)$ are not independent, so we need estimate by some decoupling tricks.

We denote $z_i := S^{-1/2}x_i$, then $z_1, \cdots, z_n \overset{\text{i.i.d.}}{\sim} \mathcal{N}(0, I_d)$ and $(x_i^\top x_j)^2 = (z_i^\top S z_j)^2$.

For any fixed $v \in \mathbb{S}^{d-1}$, by Lemma F.1, for any $\epsilon \in (0,1)$, we have

$$\mathbb{P}\left(\left|\frac{1}{n-1}\sum_{j\ne i}(v^\top z_j)^2 - 1\right| \ge \epsilon\right)$$

$$\le \mathbb{P}\left(\left|\frac{1}{n-1}\sum_{j\ne i}(v^\top z_j)^2 - 1\right| \ge \epsilon\right) \le 2\exp\left(-C_2(n-1)\epsilon^2\right),$$

where $C_2 > 0$ is an absolute constant, independent of $v$ and $\epsilon$.

Then we have

$$\mathbb{P}\left(\left|\frac{1}{n-1}\sum_{j\ne i}(x_i^\top x_j)^2 - x_i^\top S x_i\right| \ge \epsilon x_i^\top S x_i\right)$$

$$=\mathbb{P}\left(\left|\frac{1}{n-1}\sum_{j\ne i}(z_i^\top S z_j)^2 - \|S z_i\|_2^2\right| \ge \epsilon \|S z_i\|_2^2\right)$$

$$\overset{q_i := S z_i/\|S z_i\|_2}{=}\mathbb{P}\left(\left|\frac{1}{n-1}\sum_{j\ne i}(q_i^\top z_j)^2 - 1\right| \ge \epsilon\right)$$

$$=\mathbb{E}\left[\mathbb{I}\left\{\left|\frac{1}{n-1}\sum_{j\ne i}(q_i^\top z_j)^2 - 1\right| \ge 1\right\}\right]$$

$$=\mathbb{E}_{q_i}\left[\mathbb{E}\left[\mathbb{I}\left\{\left|\frac{1}{n-1}\sum_{j\ne i}(q_i^\top z_j)^2 - 1\right| \ge 1\right\}\bigg|q_i\right]\right]$$

$$\le\mathbb{E}_{q_i}\left[2\exp\left(-C_2(n-1)\epsilon^2\right)\right] = 2\exp\left(-C_2(n-1)\epsilon^2\right).$$

Applying a union bound over all $i \in [n]$, we have

$$\mathbb{P}\left(\left|\frac{1}{n-1}\sum_{j\ne i}(x_i^\top x_j)^2 - x_i^\top S x_i\right| \ge \epsilon x_i^\top S x_i, \forall i \in [n]\right) \le 2n\exp\left(-C_2(n-1)\epsilon^2\right).$$

In the other word, for any $\epsilon, \delta \in (0,1)$, if $n/\log(n/\delta) \gtrsim 1/\epsilon^2$, then $w.p.$ at least $1 - \delta/3$, we have

$$1 - \epsilon \leq \frac{\frac{1}{n-1}\sum_{j\neq i}(x_i^\top x_j)^2}{x_i^\top S x_i} \leq 1 + \epsilon, \ \forall i \in [n].$$

**Step III: Estimation of $x_i^\top S x_i$.** Let $z_i = S^{-1/2}x_i$, then $x_i^\top S x_i = z_i^\top S^2 z_i$ and $z_1, \cdots, z_n \overset{\text{i.i.d.}}{\sim} \mathcal{N}(0, I_d)$.

In the same way as Step I(i), we obtain that: for any $\epsilon, \delta \in (0,1)$, if $\text{srk}(S^2) \gtrsim \log(n)/\epsilon^2$, then $w.p.$ at least $1 - \delta/3$, we have

$$1 - \epsilon \leq \frac{x_i^\top S x_i}{\text{Tr}(S^2)} \leq 1 + \epsilon, \ \forall i \in [n].$$

Combining our results in Step I, Step II, and Step III, we obtain the result for Linear Model: for any $\epsilon, \delta \in (0,1)$, if $n/\log(n/\delta) \gtrsim 1/\epsilon^2$ and $\min\{\text{srk}(S), \text{srk}(S^2)\} \gtrsim \log(n)/\epsilon^2$, then $w.p.$ at least $1 - \delta/3 - \delta/3 - \delta/3 = 1 - \delta$, we have

$$\mu(w) \geq \frac{(1-\epsilon)^2 \text{Tr}^2(S) + (n-1)(1-\epsilon)\min_{i\in[n]} x_i^\top S x_i}{(1+\epsilon)^2 \text{Tr}^2(S) + (n-1)(1+\epsilon)\max_{i\in[n]} x_i^\top S x_i}$$

$$\geq \frac{(1-\epsilon)^2 \text{Tr}^2(S) + (n-1)(1-\epsilon)^2 \text{Tr}(S^2)}{(1+\epsilon)^2 \text{Tr}^2(S) + (n-1)(1+\epsilon)^2 \text{Tr}(S^2)} = \frac{(1-\epsilon)^2}{(1+\epsilon)^2};$$

$$\mu(w) \leq \frac{(1+\epsilon)^2 \text{Tr}^2(S) + (n-1)(1+\epsilon)\max_{i\in[n]} x_i^\top S x_i}{(1-\epsilon)^2 \text{Tr}^2(S) + (n-1)(1-\epsilon)\min_{i\in[n]} x_i^\top S x_i}$$

$$\leq \frac{(1+\epsilon)^2 \text{Tr}^2(S) + (n-1)(1+\epsilon)^2 \text{Tr}(S^2)}{(1-\epsilon)^2 \text{Tr}^2(S) + (n-1)(1-\epsilon)^2 \text{Tr}(S^2)} = \frac{(1+\epsilon)^2}{(1-\epsilon)^2}.$$

From the arbitrary of $w$, we obtain:

$$\frac{(1-\epsilon)^2}{(1+\epsilon)^2} \leq \inf_{w\in\mathbb{R}^d}\mu(w) \leq \sup_{w\in\mathbb{R}^d}\mu(w) \leq \frac{(1+\epsilon)^2}{(1-\epsilon)^2}.$$

$\square$

### B.3 Proof of Theorem 3.6

For two-layer neural networks with fixed output layer, the gradient is

$$\nabla f(x_i; \theta) = \left(a_1\sigma'(b_1^\top x_i)x_i^\top, \cdots, a_m\sigma'(b_m^\top x_i)x_i^\top\right)^\top \in \mathbb{R}^{md}.$$

For simplicity, denote $\nabla f_i(\theta) := \nabla f(x_i; \theta)$, $u_i(\theta) := f_i(\theta) - f_i(\theta^*)$. Then we have:

$$\mathcal{L}(\theta) = \frac{1}{2n}\sum_{i=1}^n u_i^2(\theta), \quad G(\theta) = \frac{1}{n}\sum_{i=1}^n \nabla f_i(\theta)\nabla f_i(\theta)^\top, \quad \Sigma_1(\theta) = \frac{1}{n}\sum_{i=1}^n u_i^2(\theta)\nabla f_i(\theta)\nabla f_i(\theta)^\top.$$

We have the following lower bound for $\mu(\theta)$:

$$\mu(\theta) = \frac{\text{Tr}\left(\left(\frac{1}{n}\sum_{i=1}^n \nabla f_i(\theta)\nabla f_i(\theta)^\top\right)\left(\frac{1}{n}\sum_{i=1}^n u_i^2(\theta)\nabla f_i(\theta)\nabla f_i(\theta)^\top\right)\right)}{\left(\frac{1}{n}\sum_{i=1}^n u_i^2(\theta)\right)\left(\frac{1}{n^2}\sum_{i=1}^n\sum_{j=1}^n (\nabla f_i(\theta)^\top\nabla f_i(\theta))^2\right)}$$

$$
\begin{aligned}
&= \frac{\frac{1}{n}\sum_{i=1}^{n} u_i^2(\theta) \frac{1}{n}\sum_{j=1}^{n} \left(\nabla f_i(\theta)^\top \nabla f_j(\theta)\right)^2}{\left(\frac{1}{n}\sum_{i=1}^{n} u_i^2(\theta)\right)\left(\frac{1}{n^2}\sum_{i=1}^{n}\sum_{j=1}^{n}\left(\nabla f_i(\theta)^\top \nabla f_i(\theta)\right)^2\right)} \\
&\geq \frac{\min_{i\in[n]} \frac{1}{n}\sum_{j=1}^{n}\left(\nabla f_i(\theta)^\top \nabla f_j(\theta)\right)^2}{\frac{1}{n^2}\sum_{i=1}^{n}\sum_{i=1}^{n}\left(\nabla f_i(\theta)^\top \nabla f_j(\theta)\right)^2} \geq \frac{\min_{i\in[n]}\frac{1}{n}\sum_{j=1}^{n}\left(\alpha^2 m x_i^\top x_j\right)^2}{\frac{1}{n^2}\sum_{i=1}^{n}\sum_{i=1}^{n}\left(\beta^2 m x_i^\top x_j\right)^2} = \frac{\alpha^2}{\beta^2}\frac{\min_{i\in[n]}\frac{1}{n}\sum_{j=1}^{n}\left(x_i^\top x_j\right)^2}{\max_{i\in[n]}\frac{1}{n}\sum_{j=1}^{n}\left(x_i^\top x_j\right)^2}.
\end{aligned}
$$

In the same way, the upper bound holds:

$$
\mu(\theta) \leq \frac{\beta^2}{\alpha^2}\frac{\max_{i\in[n]}\frac{1}{n}\sum_{j=1}^{n}\left(x_i^\top x_j\right)^2}{\min_{i\in[n]}\frac{1}{n}\sum_{j=1}^{n}\left(x_i^\top x_j\right)^2}.
$$

Notice that the terms $\dfrac{\min_{i\in[n]}\frac{1}{n}\sum_{j=1}^{n}\left(x_i^\top x_j\right)^2}{\max_{i\in[n]}\frac{1}{n}\sum_{j=1}^{n}\left(x_i^\top x_j\right)^2}$ and $\dfrac{\max_{i\in[n]}\frac{1}{n}\sum_{j=1}^{n}\left(x_i^\top x_j\right)^2}{\min_{i\in[n]}\frac{1}{n}\sum_{j=1}^{n}\left(x_i^\top x_j\right)^2}$ are independent of $\theta$ and the same as the Linear Model.

Then repeating the same proof of Theorem 3.3, the result of this theorem differs from Linear Model by only the factor $\alpha^2/\beta^2$. In other words, under the same condition with Linear Model, *w.p.* at least $1 - \delta$, we have

$$
\frac{(1-\epsilon)^2}{(1+\epsilon)^2} \leq \inf_{\theta\in\mathbb{R}^{md}} \mu(\theta) \leq \sup_{\theta\in\mathbb{R}^{md}} \mu(\theta) \leq \frac{\beta^2}{\alpha^2}\frac{(1+\epsilon)^2}{(1-\epsilon)^2}.
$$

$\square$

## C  Proofs of Section 4

For the OLM $f(x;\theta) = F(\theta)^\top x$, let $r(\theta) = F(\theta) - F(\theta^*)$ and $\nabla F(\theta) \in \mathbb{R}^{d\times p}$ the Jacobian matrix.

Then, we have for the population loss:

$$
\begin{aligned}
\bar{G}(\theta) &= \mathbb{E}\left[\nabla F(\theta)^\top x x^\top \nabla F(\theta)\right] = \nabla F(\theta)^\top S \nabla F(\theta) \\
\bar{\mathcal{L}}(\theta) &= \frac{1}{2}\mathbb{E}\left[\left(r(\theta)^\top x\right)^2\right] = \frac{1}{2}r(\theta)^\top S r(\theta) \\
\bar{\Sigma}_1(\theta) &= \mathbb{E}\left[\left(r(\theta)^\top x\right)^2 \nabla F^\top(\theta) x x^\top \nabla F(\theta)\right]
\end{aligned}
\tag{10}
$$

**Lemma C.1** (Proposition 2.3 in Wu et al. (2022)). *Let the data distribution be $\mathcal{N}(0, S)$. Then we have*

$$
\bar{\Sigma}_1(\theta) = 2\bar{\mathcal{L}}(\theta)\bar{G}(\theta) + 2\nabla\bar{\mathcal{L}}(\theta)\nabla\bar{\mathcal{L}}(\theta)^\top.
$$

**Lemma C.2.** *Under the same conditions in Lemma C.1, then we have*

$$
\left(\nabla\bar{\mathcal{L}}(\theta)^\top v\right)^2 \leq 2\bar{\mathcal{L}}(\theta) v^\top \bar{G}(\theta) v.
$$

*Proof.* For $a, b \in \mathbb{R}^p$, define $\langle a, b\rangle_S = a^\top S b$ and $\|a\|_S = \sqrt{a^\top S a}$. Since $S$ a positive semidefinite matrix, $\langle\cdot,\cdot\rangle_S$ is a well-defined inner product. Noticing that $\bar{\mathcal{L}}(\theta) = \frac{1}{2}r(\theta)^\top S r(\theta)$, we have $\nabla\bar{\mathcal{L}}(\theta) = \nabla F(\theta)^\top S r(\theta)$. Hence,

$$
\left(\nabla\bar{\mathcal{L}}(\theta)^\top v\right)^2 = v^\top \nabla F(\theta)^\top S r(\theta) r(\theta)^\top S \nabla F(\theta) v = \langle\nabla F(\theta)v, r(\theta)\rangle_S^2
$$

$$\overset{(i)}{\leq} \|\nabla F(\theta)v\|_S^2 \|r(\theta)\|_S^2 = 2\bar{\mathcal{L}}(\theta)\left(v^\top \nabla F(\theta)^\top S \nabla F(\theta)v\right) = 2\bar{\mathcal{L}}(\theta)v^\top \bar{G}(\theta)v,$$

where $(i)$ follows from Cauchy-Schwarz inequality (see Lemma F.6).

$\square$

**Lemma C.3.** *Let* $x_1, \cdots, x_n \overset{\text{i.i.d.}}{\sim} \mathcal{N}(0, I_d)$. *For any* $\epsilon, \delta \in (0, 1)$, *if we choose* $n \gtrsim (d + \log(1/\delta))/\epsilon^2$, *then* w.p. *at least* $1 - \delta$, *we have:*

$$\sup_{v \in \mathbb{S}^{d-1}} \left|\frac{1}{n}\sum_{i=1}^n (v^\top x_i)^2 - 1\right| \leq \epsilon.$$

*Proof.* By Lemma F.3, *w.p.* at least $1 - 2\exp(-u)$, we have

$$\left\|\frac{1}{n}\sum_{i=1}^n x_i x_i^\top - I_d\right\| \lesssim \sqrt{\frac{d+u}{n}} + \frac{d+u}{n}.$$

Equivalently, we can rewrite this conclusion. For any $\epsilon, \delta \in (0, 1)$, if we choose $n \gtrsim (d + \log(1/\delta))/\epsilon^2$, then *w.p.* at least $1 - \delta$, we have:

$$\sup_{v \in \mathbb{S}^{d-1}} \left|\frac{1}{n}\sum_{i=1}^n (v^\top x_i)^2 - 1\right| \leq \left\|\frac{1}{n}\sum_{i=1}^n x_i x_i^\top - I_d\right\| \leq \epsilon.$$

$\square$

**Lemma C.4** (Lemma 17 in (Cai et al., 2022)). *For any* $0 < \epsilon < 1/2$, *there are constants* $C_1 = C_1(\epsilon) > 0$ *and* $C_2 = C_2(\epsilon) > 0$, *such that if* $n \geq C_1 d\log d$, *then with probability at least* $1 - \frac{C_2}{n^2}$, *it holds*

$$\sup_{u \in \mathbb{S}^{d-1}} \left|\frac{1}{n}\sum_{i=1}^n (x_i^\top u)^2 (x_i^\top v)^2 - \mathbb{E}\left[(x^\top u)^2 (x^\top v)^2\right]\right| \leq \epsilon.$$

## C.1 Proof of Theorem 4.2

We first need a few lemmas.

**Lemma C.5.** *Let* $z_1, \cdots, z_n \overset{\text{i.i.d.}}{\sim} \mathcal{N}(0, I_d)$. *If* $n \gtrsim d^2 + \log^2(1/\delta)$, *then* w.p. *at least* $1 - \delta$, *we have*

$$\sup_{v \in \mathbb{S}^{d-1}} \frac{1}{n}\sum_{i=1}^n (z_i^\top v)^4 \leq 8.$$

*Proof.* For $\mathbb{S}^{d-1}$, its covering number has the bound:

$$\left(\frac{1}{\rho}\right)^d \leq \mathcal{N}(\mathbb{S}^{d-1}, \rho) \leq \left(\frac{2}{\rho} + 1\right)^d,$$

so there exist a $\rho$-net on $\mathbb{S}^{d-1}$: $\mathcal{V} \subset \mathbb{S}^{d-1}$, s.t. $|\mathcal{V}| \leq \left(\frac{2}{\rho} + 1\right)^d$.

**Step I: Bounding the term on the $\rho$-net.** For a fixed $v \in \mathcal{V}$, due to $z_i \overset{\text{i.i.d.}}{\sim} \mathcal{N}(0, I_d)$, we can verify $(z_i^\top v)^4$ is sub-Weibull random variable:

$$\mathbb{E}\exp\left(\left((z_i^\top v)^4\right)^{1/2}\right) = \mathbb{E}\exp\left((z_i^\top v)^2\right) \lesssim 1,$$

which means that there exist an absolute constant $C_1 \geq 1$ s.t. $\left\|(z_i^\top v)^4\right\|_{\psi_{1/2}} \leq C_1$.

By the concentration inequality for Sub-Weibull distribution with $\beta = 1/2$ (Lemma F.5) and $\mathbb{E}\left[(y^\top v)^4\right] = 3$, there exists an absolute constant $C_2 \geq 1$ s.t.

$$\mathbb{P}\left(\left|\frac{1}{n}\sum_{i=1}^n \left[(z_i^\top v)^4\right] - 3\right| > \phi(n;\delta)\right) \leq 2\delta,$$

where $\phi(n;\delta) = C_2(\sqrt{\frac{\log(1/\delta)}{n}} + \frac{\log^2(1/\delta)}{n})$. Applying a union bound over $v \in \mathcal{V}$, we have:

$$\mathbb{P}\left(\exists v \in \mathcal{V} \, s.t. \left|\frac{1}{n}\sum_{i=1}^n \left[(z_i^\top v)^4\right] - 3\right| > \phi(n;\delta)\right)$$

$$\leq \mathbb{P}\left(\bigcup_{v \in \mathcal{V}} \left\{\left|\frac{1}{n}\sum_{i=1}^n \left[(z_i^\top v)^4\right] - 3\right| > \phi(n;\delta)\right\}\right) \leq \sum_{v \in \mathcal{V}} \mathbb{P}\left(\left|\frac{1}{n}\sum_{i=1}^n \left[(z_i^\top v)^4\right] - 3\right| > \phi(n;\delta)\right)$$

$$\leq 2|\mathcal{V}| \exp\left(-\frac{n}{C_2^2}\right) = 2\left(\frac{2}{\rho} + 1\right)^d \delta.$$

So *w.p.* at least $1 - 2\left(\frac{2}{\rho} + 1\right)^d \delta$, we have:

$$\max_{v \in \mathcal{V}} \frac{1}{n}\sum_{i=1}^n \left[(z_i^\top v)^4\right] \leq 3 + \phi(n;\delta).$$

**Step II: Estimate the error of the $\rho$-net approximation.** For simplicity, we denote

$$P := \max_{v \in \mathbb{S}^{d-1}} \frac{1}{n}\sum_{i=1}^n \left[(z_i^\top v)^4\right], \quad Q := \max_{v \in \mathcal{V}} \frac{1}{n}\sum_{i=1}^n \left[(z_i^\top v)^4\right].$$

Let $v \in \mathbb{S}^{d-1}$ such that $\frac{1}{n}\sum_{i=1}^n \left[(z_i^\top v)^4\right] = P$, then there exist $v_0 \in \mathcal{V}$, s.t. $\|v - v_0\| \leq \rho$.

On the one hand,

$$\left|\frac{1}{n}\sum_{i=1}^n (z_i^\top v)^4 - \frac{1}{n}\sum_{i=1}^n (z_i^\top v_0)^4\right| = \left|\frac{1}{n}\sum_{i=1}^n \left((z_i^\top v)^4 - (z_i^\top v_0)^4\right)\right|$$

$$= \left|\frac{1}{n}\sum_{i=1}^n \left(z_i^\top (v - v_0)\right)\left(z_i^\top (v + v_0)\right)\left((z_i^\top v)^2 + (z_i^\top v_0)^2\right)\right|$$

$$\leq \left|\frac{1}{n}\sum_{i=1}^n \left(z_i^\top (v - v_0)\right)\left(z_i^\top (v + v_0)\right)(z_i^\top v)^2\right| + \left|\frac{1}{n}\sum_{i=1}^n \left(z_i^\top (v - v_0)\right)\left(z_i^\top (v + v_0)\right)(z_i^\top v_0)^2\right|$$

$$\leq \sqrt{\frac{1}{n}\sum_{i=1}^n \left(z_i^\top (v - v_0)\right)^2 \left(z_i^\top (v + v_0)\right)^2}\left(\sqrt{\frac{1}{n}\sum_{i=1}^n (z_i^\top v)^4} + \sqrt{\frac{1}{n}\sum_{i=1}^n (z_i^\top v_0)^4}\right)$$

$$\leq \sqrt[4]{\frac{1}{n}\sum_{i=1}^n \left(z_i^\top (v - v_0)\right)^4} \sqrt[4]{\frac{1}{n}\sum_{i=1}^n \left(z_i^\top (v + v_0)\right)^4}\left(\sqrt{\frac{1}{n}\sum_{i=1}^n (z_i^\top v)^4} + \sqrt{\frac{1}{n}\sum_{i=1}^n (z_i^\top v_0)^4}\right)$$

$$\leq \|v - v_0\| P^{1/4} \|v + v_0\| P^{1/4} (\sqrt{P} + \sqrt{Q}) \leq 2\rho\sqrt{P}(\sqrt{P} + \sqrt{Q})$$

On the other hand,

$$\left|\frac{1}{n}\sum_{i=1}^n (z_i^\top v)^4 - \frac{1}{n}\sum_{i=1}^n (z_i^\top v_0)^4\right| \geq P - \sum_{i=1}^n (z_i^\top v_0)^4 \geq P - Q.$$

Hence, we obtain

$$P - Q \leq 2\rho\sqrt{P}(\sqrt{P} + \sqrt{Q}),$$

which means that

$$P \leq \left(\frac{1}{1 - 2\rho}\right)^2 Q.$$

**Step III: The bound for any $v \in \mathbb{S}^{d-1}$.** Select $\rho = \frac{1}{2}(1 - \frac{1}{\sqrt{2}})$ and denote $\delta' = 2(\frac{2}{\rho} + 1)^d \delta$. And we choose $n \gtrsim d^2 + \log^2(1/\delta')$, which ensures $\phi(n; \delta) \leq 1$.

Then combining the results in Step I and Step II, we know that: *w.p.* at least $1 - \delta'$, we have:

$$\max_{v \in \mathcal{V}} \frac{1}{n} \sum_{i=1}^n \left[(z_i^\top v)^4\right] \leq 3 + 1 = 4; \quad \max_{v \in \mathbb{S}^{d-1}} \frac{1}{n} \sum_{i=1}^n \left[(z_i^\top v)^4\right] \leq 2 \max_{v \in \mathcal{V}} \frac{1}{n} \sum_{i=1}^n \left[(z_i^\top v)^4\right],$$

which means

$$\max_{v \in \mathbb{S}^{d-1}} \frac{1}{n} \sum_{i=1}^n \left[(z_i^\top v)^4\right] \leq 2 \cdot 4 = 8.$$

$\square$

**Lemma C.6.** *Let $x_1, \cdots, x_n \overset{\text{i.i.d.}}{\sim} \mathcal{N}(0, I_d)$. For any $\epsilon, \delta \in (0, 1)$, if we choose*

$$n \gtrsim \max\left\{\left(d^2 \log^2(1/\epsilon) + \log^2(1/\delta)\right)/\epsilon, \left(d \log(1/\epsilon) + \log(1/\delta)\right)/\epsilon^2\right\},$$

*then* w.p. *at least $1 - \delta$, we have:*

$$\sup_{w,v \in \mathbb{S}^{d-1}} \left|\frac{1}{n} \sum_{i=1}^n (w^\top x_i)^2 (v^\top x_i)^2 - \mathbb{E}\left[(w^\top x_1)^2 (v^\top x_1)^2\right]\right| \leq \epsilon.$$

*Proof.* For $\mathbb{S}^{d-1}$, its covering number has the bound:

$$\left(\frac{1}{\rho}\right)^d \leq \mathcal{N}(\mathbb{S}^{d-1}, \rho) \leq \left(\frac{2}{\rho} + 1\right)^d,$$

so there exist two $\rho$-nets on $\mathbb{S}^{d-1}$: $\mathcal{W} \subset \mathbb{S}^{d-1}$ and $\mathcal{V} \subset \mathbb{S}^{d-1}$, s.t.

$$|\mathcal{W}| \leq \left(\frac{2}{\rho} + 1\right)^d, \quad |\mathcal{V}| \leq \left(\frac{2}{\rho} + 1\right)^d.$$

**Step I: Bounding the term on the $\rho$-net.** In this step, will estimate the term

$$\left|\frac{1}{n} \sum_{i=1}^n (w^\top x_i)^2 (v^\top x_i)^2 - \mathbb{E}\left[(w^\top x)^2 (v^\top x)^2\right]\right|$$

for any $w \in \mathcal{W}$ and $v \in \mathcal{V}$.

For fixed $w \in \mathcal{W}$ and $v \in \mathcal{V}$, we denote $X_i^{w,v} := (w^\top x_i)^2 (v^\top x_i)^2$. We can verify $X_i$ is a sub-Weibull random variable with $\beta = 1/2$ (Definition F.4):

$$\mathbb{E}\left[\exp\left(\left|(w^\top x_i)^2 (v^\top x_i)^2\right|^{1/2}\right)\right] = \mathbb{E}\left[\exp\left(|w^\top x_i||v^\top x_i|\right)\right]$$

$$\leq \mathbb{E}\left[\exp\left(\frac{(w^\top x_i)^2 + (v^\top x_i)^2}{2}\right)\right] = \mathbb{E}\left[\exp\left(\frac{(w^\top x_i)^2}{2}\right)\exp\left(\frac{(v^\top x_i)^2}{2}\right)\right]$$

$$\overset{\text{Lemma F.6}}{\leq} \sqrt{\mathbb{E}\left[\exp\left((w^\top x_i)^2\right)\right]} \cdot \sqrt{\mathbb{E}\left[\exp\left((v^\top x_i)^2\right)\right]} \overset{\left\|(v^\top x_i)^2\right\|_{\psi_1} \leq C_3}{\lesssim} 1,$$

which means that there exists an absolute constant $C_4 \geq 1$, s.t. $\|X_i^{w,v}\|_{\psi_{1/2}} \leq C_4$. By the concentration inequality for Sub-Weibull distribution with $\beta = 1/2$ (Lemma F.5), there exists an absolute constant $C_5 \geq 1$, s.t.

$$\mathbb{P}\left(\left|\frac{1}{n}\sum_{i=1}^n X_i^{w,v} - \frac{1}{n}\sum_{i=1}^n \mathbb{E}[X_i^{w,v}]\right| > \psi(n;\delta)\right) \leq \delta.$$

where $\psi(n;\delta) = C_5\left(\sqrt{\frac{\log(1/\delta)}{n}} + \frac{(\log(1/\delta))^2}{n}\right)$.

Applying an union bound over $w \in \mathcal{W}$ and $v \in \mathcal{V}$, we have:

$$\mathbb{P}\left(\exists w \in \mathcal{W}, v \in \mathcal{V}, \text{s.t.}\left|\frac{1}{n}\sum_{i=1}^n X_i^{w,v} - \frac{1}{n}\sum_{i=1}^n \mathbb{E}[X_i^{w,v}]\right| > \psi(n;\delta)\right)$$

$$\leq \mathbb{P}\left(\bigcup_{(w,v)\in\mathcal{W}\times\mathcal{V}}\left\{\exists w \in \mathcal{W}, v \in \mathcal{V}, \text{s.t.}\left|\frac{1}{n}\sum_{i=1}^n X_i^{w,v} - \frac{1}{n}\sum_{i=1}^n \mathbb{E}[X_i^{w,v}]\right| > \psi(n;\delta)\right\}\right)$$

$$\leq \sum_{(w,v)\in\mathcal{W}\times\mathcal{V}} \mathbb{P}\left(\exists w \in \mathcal{W}, v \in \mathcal{V}, \text{s.t.}\left|\frac{1}{n}\sum_{i=1}^n X_i^{w,v} - \frac{1}{n}\sum_{i=1}^n \mathbb{E}[X_i^{w,v}]\right| > \psi(n;\delta)\right)$$

$$\leq 2|\mathcal{W}||\mathcal{V}|\delta \leq 2\left(\frac{2}{\rho}+1\right)^{2d}\delta.$$

So *w.p.* at least $1 - 2\left(\frac{2}{\rho}+1\right)^{2d}\delta$, we have:

$$\sup_{w\in\mathcal{W},v\in\mathcal{V}}\left|\frac{1}{n}\sum_{i=1}^n (w^\top x_i)^2(v^\top x_i)^2 - \mathbb{E}\left[(w^\top x)^2(v^\top x)^2\right]\right| \leq \psi(n;\delta).$$

**Step II: Estimate the population error of the $\rho$-net approximation.** Let $w, v, w_0, v_0 \in \mathbb{S}^{d-1}$, s.t. $\|w - w_0\| \leq \rho$ and $\|v - v_0\| \leq \rho$. For the population error, we have

$$\left|\mathbb{E}\left[(w^\top x)^2(v^\top x)^2\right] - \mathbb{E}\left[(w_0^\top x)^2(v_0^\top x)^2\right]\right|$$

$$= \left|\mathbb{E}\left[\left((w^\top x)^2 - (w_0^\top x)^2\right)(v^\top x)^2\right] + \mathbb{E}\left[(w_0^\top x)^2\left((v^\top x)^2 - (v_0^\top x)^2\right)\right]\right|$$

$$\leq \left|\mathbb{E}\left[\left((w^\top x)^2 - (w_0^\top x)^2\right)(v^\top x)^2\right]\right| + \left|\mathbb{E}\left[(w_0^\top x)^2\left((v^\top x)^2 - (v_0^\top x)^2\right)\right]\right|$$

We first bound $\left|\mathbb{E}\left[\left((w^\top x)^2 - (w_0^\top x)^2\right)(v^\top x)^2\right]\right|$:

$$\left|\mathbb{E}\left[\left((w^\top x)^2 - (w_0^\top x)^2\right)(v^\top x)^2\right]\right| = \left|\mathbb{E}\left[\left((w - w_0)^\top x x^\top(w + w_0)(v^\top x)^2\right]\right|$$

$$\leq \left(\mathbb{E}\left[\left((w - w_0)^\top x x^\top(w + w_0)\right)^2\right]\right)^{1/2}\left(\mathbb{E}\left[(v^\top x)^4\right]\right)^{1/2}$$

$$\leq \left(\mathbb{E}\left[\left((w - w_0)^\top x\right)^4\right]\right)^{1/4}\left(\mathbb{E}\left[\left((w + w_0)^\top x\right)^4\right]\right)^{1/4}\left(\mathbb{E}\left[(v^\top x)^4\right]\right)^{1/2}$$

$$\leq 3 \left\| (w - w_0) \right\| \left\| (w + w_0) \right\| \left\| v \right\|^2 \leq 6\rho.$$

Repeating the proof above, we also have:

$$\left| \mathbb{E}\left[ \left( (w^\top x)^2 - (w_0^\top x)^2 \right) (v^\top x)^2 \right] \right| \leq 6\rho.$$

Combining these two inequalities, we have:

$$\left| \mathbb{E}\left[ (w^\top x)^2 (v^\top x)^2 \right] - \mathbb{E}\left[ (w_0^\top x)^2 (v_0^\top x)^2 \right] \right| \leq 6\rho + 6\rho = 12\rho.$$

Due to the arbitrariness of $w, v, w_0, v_0$, we obtain

$$\sup_{\substack{w,v,w_0,v_0 \in \mathbb{S}^{d-1} \\ \|w-w_0\| \leq \rho, \|v-v_0\| \leq \rho}} \left| \mathbb{E}\left[ (w^\top x)^2 (v^\top x)^2 \right] - \mathbb{E}\left[ (w_0^\top x)^2 (v_0^\top x)^2 \right] \right| \leq 12\rho.$$

**Step III: Estimate the empirical error of the $\rho$-net approximation.** Let $w, v, w_0, v_0 \in \mathbb{S}^{d-1}$, s.t. $\|w - w_0\| \leq \rho$ and $\|v - v_0\| \leq \rho$. For the empirical error, we have

$$\left| \frac{1}{n} \sum_{i=1}^{n} (w^\top x_i)^2 (v^\top x_i)^2 - \frac{1}{n} \sum_{i=1}^{n} (w_0^\top x_i)^2 (v_0^\top x_i)^2 \right|$$

$$= \left| \frac{1}{n} \sum_{i=1}^{n} \left[ \left( (w^\top x_i)^2 - (w_0^\top x_i)^2 \right) (v^\top x_i)^2 \right] + \frac{1}{n} \sum_{i=1}^{n} \left[ (w_0^\top x_i)^2 \left( (v^\top x_i)^2 - (v_0^\top x_i)^2 \right) \right] \right|$$

$$\leq \left| \frac{1}{n} \sum_{i=1}^{n} \left[ \left( (w^\top x_i)^2 - (w_0^\top x_i)^2 \right) (v^\top x_i)^2 \right] \right| + \left| \frac{1}{n} \sum_{i=1}^{n} \left[ (w_0^\top x_i)^2 \left( (v^\top x_i)^2 - (v_0^\top x_i)^2 \right) \right] \right|$$

We first bound $\left| \frac{1}{n} \sum_{i=1}^{n} \left[ \left( (w^\top x_i)^2 - (w_0^\top x_i)^2 \right) (v^\top x_i)^2 \right] \right|$:

$$\left| \frac{1}{n} \sum_{i=1}^{n} \left[ \left( (w^\top x_i)^2 - (w_0^\top x_i)^2 \right) (v^\top x_i)^2 \right] \right| = \left| \frac{1}{n} \sum_{i=1}^{n} \left[ \left( (w - w_0)^\top x_i x_i^\top (w + w_0)(v^\top x_i)^2 \right) \right] \right|$$

$$\leq 2\rho \sup_{u \in \mathbb{S}^{d-1}} \frac{1}{n} \sum_{i=1}^{n} (x_i^\top u)^4.$$

Repeating the proof above, we also have $\left| \frac{1}{n} \sum_{i=1}^{n} \left[ (w_0^\top x_i)^2 \left( (v^\top x_i)^2 - (v_0^\top x_i)^2 \right) \right] \right| \leq 2\rho \sup_{u \in \mathbb{S}^{d-1}} \frac{1}{n} \sum_{i=1}^{n} (x_i^\top u)^4.$

Combining these two bounds, we have:

$$\left| \frac{1}{n} \sum_{i=1}^{n} (w^\top x_i)^2 (v^\top x_i)^2 - \frac{1}{n} \sum_{i=1}^{n} (w_0^\top x_i)^2 (v_0^\top x_i)^2 \right| \leq 4\rho \sup_{u \in \mathbb{S}^{d-1}} \frac{1}{n} \sum_{i=1}^{n} (x_i^\top u)^4.$$

Using Lemma C.5, if $n \gtrsim d^2 + \log^2(1/\delta')$, then *w.p.* at least $1 - \delta'/2$, we have $\sup_{u \in \mathbb{S}^{d-1}} \frac{1}{n} \sum_{i=1}^{n} (x_i^\top u)^4 \leq 8$.

Hence, *w.p.* at least $1 - \delta'/2$, we have

$$\left| \frac{1}{n} \sum_{i=1}^{n} (w^\top x_i)^2 (v^\top x_i)^2 - \frac{1}{n} \sum_{i=1}^{n} (w_0^\top x_i)^2 (v_0^\top x_i)^2 \right| \leq 32\rho.$$

Due to the arbitrariness of $w, v, w_0, v_0$, we obtain

$$\sup_{\substack{w,v,w_0,v_0 \in \mathbb{S}^{d-1} \\ \|w-w_0\| \leq \rho, \|v-v_0\| \leq \rho}} \left| \frac{1}{n} \sum_{i=1}^{n} (w^\top x_i)^2 (v^\top x_i)^2 - \frac{1}{n} \sum_{i=1}^{n} (w_0^\top x_i)^2 (v_0^\top x_i)^2 \right| \leq 32\rho.$$

**Step IV: The bound for any** $w, v \in \mathbb{S}^{d-1}$**.** Combining the results in Step I, II, and II, we know that *w.p.* at least $1 - \frac{\delta'}{2} - (\frac{2}{\rho} + 1)^d$, we have

$$\sup_{w \in \mathcal{W}, v \in \mathcal{V}} \left| \frac{1}{n} \sum_{i=1}^{n} (w^\top x_i)^2 (v^\top x_i)^2 - \mathbb{E}\left[ (w^\top x)^2 (v^\top x)^2 \right] \right| \leq \psi(n; \delta),$$

$$\sup_{\substack{w, v, w_0, v_0 \in \mathbb{S}^{d-1} \\ \|w - w_0\| \leq \rho, \|v - v_0\| \leq \rho}} \left| \mathbb{E}\left[ (w^\top x)^2 (v^\top x)^2 \right] - \mathbb{E}\left[ (w_0^\top x)^2 (v_0^\top x)^2 \right] \right| \leq 12\rho,$$

$$\sup_{\substack{w, v, w_0, v_0 \in \mathbb{S}^{d-1} \\ \|w - w_0\| \leq \rho, \|v - v_0\| \leq \rho}} \left| \frac{1}{n} \sum_{i=1}^{n} (w^\top x_i)^2 (v^\top x_i)^2 - \frac{1}{n} \sum_{i=1}^{n} (w_0^\top x_i)^2 (v_0^\top x_i)^2 \right| \leq 32\rho.$$

Then for any $w, v \in \mathbb{S}^{d-1}$, there exists $w_0 \in \mathcal{W}, v_0 \in \mathcal{V}$ s.t. $\|w - w_0\| \leq \rho$ and $\|v - v_0\| \leq \rho$, so

$$\left| \frac{1}{n} \sum_{i=1}^{n} (w^\top x_i)^2 (v^\top x_i)^2 - \mathbb{E}\left[ (w^\top x)^2 (v^\top x)^2 \right] \right|$$

$$= \left| \frac{1}{n} \sum_{i=1}^{n} (w^\top x_i)^2 (v^\top x_i)^2 - \frac{1}{n} \sum_{i=1}^{n} (w_0^\top x_i)^2 (v_0^\top x_i)^2 + \frac{1}{n} \sum_{i=1}^{n} (w_0^\top x_i)^2 (v_0^\top x_i)^2 \right.$$

$$\left. - \mathbb{E}\left[ (w_0^\top x)^2 (v_0^\top x)^2 \right] + \mathbb{E}\left[ (w_0^\top x)^2 (v_0^\top x)^2 \right] - \mathbb{E}\left[ (w^\top x)^2 (v^\top x)^2 \right] \right|$$

$$\leq \left| \frac{1}{n} \sum_{i=1}^{n} (w^\top x_i)^2 (v^\top x_i)^2 - \frac{1}{n} \sum_{i=1}^{n} (w_0^\top x_i)^2 (v_0^\top x_i)^2 \right|$$

$$+ \left| \frac{1}{n} \sum_{i=1}^{n} (w_0^\top x_i)^2 (v_0^\top x_i)^2 - \mathbb{E}\left[ (w_0^\top x)^2 (v_0^\top x)^2 \right] \right| + \left| \mathbb{E}\left[ (w_0^\top x)^2 (v_0^\top x)^2 \right] - \mathbb{E}\left[ (w^\top x)^2 (v^\top x)^2 \right] \right|$$

$$\leq \sup_{\substack{w, v, w_0, v_0 \in \mathbb{S}^{d-1} \\ \|w - w_0\| \leq \rho, \|v - v_0\| \leq \rho}} \left| \frac{1}{n} \sum_{i=1}^{n} (w^\top x_i)^2 (v^\top x_i)^2 - \frac{1}{n} \sum_{i=1}^{n} (w_0^\top x_i)^2 (v_0^\top x_i)^2 \right|$$

$$+ \sup_{w \in \mathcal{W}, v \in \mathcal{V}} \left| \frac{1}{n} \sum_{i=1}^{n} (w^\top x_i)^2 (v^\top x_i)^2 - \mathbb{E}\left[ (w^\top x)^2 (v^\top x)^2 \right] \right|$$

$$+ \sup_{\substack{w, v, w_0, v_0 \in \mathbb{S}^{d-1} \\ \|w - w_0\| \leq \rho, \|v - v_0\| \leq \rho}} \left| \mathbb{E}\left[ (w^\top x)^2 (v^\top x)^2 \right] - \mathbb{E}\left[ (w_0^\top x)^2 (v_0^\top x)^2 \right] \right|$$

$$\leq 32\rho + \psi(n; \delta) + 12\rho = 44\rho + \psi(n; \delta).$$

Due to the arbitrariness of $w, v$, we have

$$\sup_{w, v \in \mathbb{S}^{d-1}} \left| \frac{1}{n} \sum_{i=1}^{n} (w^\top x_i)^2 (v^\top x_i)^2 - \mathbb{E}\left[ (w^\top x)^2 (v^\top x)^2 \right] \right| \leq 44\rho + \psi(n; \delta)$$

Select $\rho = \frac{\epsilon}{66}$ and $\delta'/2 = 2(1 + \frac{2}{\rho})^{2d}\delta$. And we choose

$$n \gtrsim \max \left\{ \left( d^2 \log^2(1/\epsilon) + \log^2(1/\delta) \right) / \epsilon, \left( d \log(1/\epsilon) + \log(1/\delta) \right) / \epsilon^2 \right\},$$

which satisfies $\psi(n; \delta) \leq \epsilon/3$.

Then *w.p.* at least $1 - \delta'/2 - \delta'/2 = 1 - \delta'$, we have

$$\sup_{w, v \in \mathbb{S}^{d-1}} \left| \frac{1}{n} \sum_{i=1}^{n} (w^\top x_i)^2 (v^\top x_i)^2 - \mathbb{E}\left[ (w^\top x)^2 (v^\top x)^2 \right] \right| \leq \frac{44}{66}\epsilon + \frac{1}{3}\epsilon = \epsilon.$$

$\square$

With the preparation of Lemma C.1, C.3, and C.6, now we give the proof of Theorem 4.2.

**Proof of Theorem 4.2.** Let $z_i = S^{-1/2}x_i$. Then $z_1, \cdots, z_n \overset{\text{i.i.d.}}{\sim} \mathcal{N}(0, I_d)$.

$$g(\theta; v) = \frac{\frac{1}{n}\sum_{i=1}^n \left(r(\theta)^\top x_i\right)^2 \left((\nabla F(\theta)v)^\top x_i\right)^2}{\frac{1}{n}\sum_{i=1}^n \left(r(\theta)^\top x_i\right)^2 \cdot \frac{1}{n}\sum_{i=1}^n \left((\nabla F(\theta)v)^\top x_i\right)^2}$$

$$= \frac{\frac{1}{n}\sum_{i=1}^n \left((S^{1/2}r(\theta))^\top z_i\right)^2 \left((S^{1/2}\nabla F(\theta)v)^\top z_i\right)^2}{\frac{1}{n}\sum_{i=1}^n \left((S^{1/2}r(\theta))^\top z_i\right)^2 \cdot \frac{1}{n}\sum_{i=1}^n \left((S^{1/2}\nabla F(\theta)v)^\top z_i\right)^2},$$

Case (i). If $S^{1/2}r(\theta) = 0$ or $S^{1/2}\nabla F(\theta)v = 0$, we have $g(\theta; v) = \frac{0}{0} = 1$, this theorem holds.

Case (ii). If $S^{1/2}r(\theta) \neq 0$ and $S^{1/2}\nabla F(\theta)v \neq 0$, we define the following normalized vectors:

$$\tilde{r}(\theta) := \frac{S^{1/2}r(\theta)}{\|S^{1/2}r(\theta)\|} \in \mathbb{S}^{d-1} \quad \tilde{w}(\theta; v) := \frac{S^{1/2}\nabla F(\theta)v}{\|S^{1/2}\nabla F(\theta)v\|} \in \mathbb{S}^{d-1}.$$

From the homogeneity of $g(\theta; v)$, we have:

$$g(\theta; v) = \frac{\frac{1}{n}\sum_{i=1}^n \left(\tilde{r}(\theta)^\top z_i\right)^2 \left(\tilde{w}(\theta; v)^\top z_i\right)^2}{\frac{1}{n}\sum_{i=1}^n \left(\tilde{r}(\theta)^\top z_i\right)^2 \cdot \frac{1}{n}\sum_{i=1}^n \left(\tilde{w}(\theta; v)^\top z_i\right)^2}.$$

By Lemma C.3 and C.6, for any $\epsilon, \delta \in (0, 1)$, if we choose

$$n \gtrsim \max \left\{ \left(d^2 \log^2(1/\epsilon) + \log^2(1/\delta)\right)/\epsilon, (d\log(1/\epsilon) + \log(1/\delta))/\epsilon^2 \right\},$$

then $w.p.$ at least $1 - \delta$, the following inequalities hold:

$$\sup_{v \in \mathbb{S}^{d-1}} \left| \frac{1}{n}\sum_{i=1}^n (v^\top z_i)^2 - 1 \right| \leq \epsilon,$$

$$\sup_{w, v \in \mathbb{S}^{d-1}} \left| \frac{1}{n}\sum_{i=1}^n (w^\top z_i)^2(v^\top z_i)^2 - \mathbb{E}\left[(w^\top z_1)^2(v^\top z_1)^2\right] \right| \leq \epsilon;$$

These imply that for any $\theta, v \in \mathbb{R}^p$, we have:

$$\frac{\mathbb{E}\left[(\tilde{r}(\theta)^\top y)^2(\tilde{w}(\theta; v)^\top y)^2\right] - \epsilon}{(1 + \epsilon)^2} \leq g(\theta; v) \leq \frac{\mathbb{E}\left[(\tilde{r}(\theta)^\top z_1)^2(\tilde{w}(\theta; v)^\top z_1)^2\right] + \epsilon}{(1 - \epsilon)^2}. \tag{11}$$

First, we derive the upper bound for (11):

$$\text{RHS} = \frac{\epsilon}{(1 - \epsilon)^2} + \frac{\mathbb{E}\left[(\tilde{r}(\theta)^\top y)^2(\tilde{w}(\theta; v)^\top y)^2\right]}{(1 - \epsilon)^2 \left(\tilde{r}(\theta)^\top \tilde{r}(\theta)\right)\left(\tilde{w}(\theta; v)^\top \tilde{w}(\theta; v)\right)}$$

$$\overset{\text{Homogeneity}}{=} \frac{\epsilon}{(1 - \epsilon)^2} + \frac{\mathbb{E}\left[((S^{1/2}r(\theta))^\top y)^2((S^{1/2}\nabla F(\theta)v)^\top y)^2\right]}{(1 - \epsilon)^2 \left((S^{1/2}r(\theta))^\top S^{1/2}r(\theta)\right)\left((S^{1/2}\nabla F(\theta)v)^\top (S^{1/2}\nabla F(\theta)v)\right)}$$

$$
\begin{aligned}
=&\frac{\epsilon}{(1-\epsilon)^2}+\frac{\mathbb{E}\left[(r(\theta)^\top x)^2\left(v^\top\nabla F(\theta)^\top xx^\top\nabla F(\theta)v\right)\right]}{(1-\epsilon)^2\left(r(\theta)^\top Sr(\theta)\right)\left(v^\top\nabla F(\theta)^\top S\nabla F(\theta)v\right)}\\
\overset{(10)}{=}&\frac{\epsilon}{(1-\epsilon)^2}+\frac{v^\top\bar\Sigma_1(\theta)v}{2(1-\epsilon)^2\bar{\mathcal{L}}(\theta)v^\top\bar G(\theta)v}\overset{\text{Lemma C.1}}{=}\frac{\epsilon}{(1-\epsilon)^2}+\frac{2\bar{\mathcal{L}}(\theta)v^\top\bar G(\theta)v+2\left(\nabla\bar{\mathcal{L}}(\theta)^\top v\right)^2}{2(1-\epsilon)^2\bar{\mathcal{L}}(\theta)v^\top\bar G(\theta)v}\\
=&\frac{1+\epsilon}{(1-\epsilon)^2}+\frac{\left(\nabla\bar{\mathcal{L}}(\theta)^\top v\right)^2}{(1-\epsilon)^2\bar{\mathcal{L}}(\theta)v^\top\bar G(\theta)v}\overset{\text{Lemma C.2}}{\leq}\frac{1+\epsilon}{(1-\epsilon)^2}+\frac{2}{(1-\epsilon)^2}=\frac{3+\epsilon}{(1-\epsilon)^2}.
\end{aligned}
$$

Moreover, if $\langle v,\mathcal{L}(\theta)\rangle=0$, then the bound is

$$
\text{RHS}\leq\frac{3+\epsilon}{(1-\epsilon)^2}.
$$

In the similar way, we can derive the lower bound for (11):

$$
\begin{aligned}
\text{LHS}=&\frac{v^\top\bar\Sigma_1(\theta)v}{2(1+\epsilon)^2\bar{\mathcal{L}}(\theta)v^\top\bar G(\theta)v}-\frac{\epsilon}{(1+\epsilon)^2}\overset{\text{Lemma C.1}}{=}\frac{2\bar{\mathcal{L}}(\theta)v^\top\bar G(\theta)v+\left(\nabla\bar{\mathcal{L}}(\theta)^\top v\right)^2}{2(1+\epsilon)^2\bar{\mathcal{L}}(\theta)v^\top\bar G(\theta)v}-\frac{\epsilon}{(1+\epsilon)^2}\\
\geq&\frac{1}{(1+\epsilon)^2}-\frac{\epsilon}{(1+\epsilon)^2}=\frac{1-\epsilon}{(1+\epsilon)^2}.
\end{aligned}
$$

So for any $S^{1/2}u(\theta)\neq 0, S^{1/2}\nabla F(\theta)v\neq 0$, we have

$$
\frac{1-\epsilon}{(1+\epsilon)^2}\leq g(\theta;v)\leq\frac{3+\epsilon}{(1-\epsilon)^2}.
$$

Hence, we have proved this theorem: For any $\epsilon,\delta>0$, if

$$
n\gtrsim\max\left\{\left(d^2\log^2\left(1/\epsilon\right)+\log^2(1/\delta)\right)/\epsilon,\left(d\log\left(1/\epsilon\right)+\log(1/\delta)\right)/\epsilon^2\right\},
$$

then w.p. at least $1-\delta$, the strong alignment holds uniformly:

$$
\frac{1-\epsilon}{(1+\epsilon)^2}\leq\inf_{\theta,v\in\mathbb{R}^p}g(\theta;v)\leq\sup_{\theta,v\in\mathbb{R}^p}g(\theta;v)\leq\frac{3+\epsilon}{(1-\epsilon)^2},
$$

$\square$

## C.2 Proof of Theorem 4.3

With the preparation of Lemma C.3 and Lemma C.4, now we give the proof of Theorem 4.3.

We follow the proof of Theorem 4.2. Notice that for linear model,

$$
\tilde w(\theta,v)=\frac{S^{1/2}\nabla F(\theta)v}{\left\|S^{1/2}\nabla F(\theta)v\right\|}=\frac{S^{1/2}v}{\left\|S^{1/2}v\right\|},
$$

independent of $\theta$. Thus, we denote it as $\tilde w(v):=\frac{S^{1/2}v}{\left\|S^{1/2}v\right\|}$. Then,

$$
g(\theta;v)=\frac{\frac{1}{n}\sum_{i=1}^n\left(\tilde r(\theta)^\top z_i\right)^2\left(\tilde w(v)^\top z_i\right)^2}{\frac{1}{n}\sum_{i=1}^n\left(\tilde r(\theta)^\top z_i\right)^2\cdot\frac{1}{n}\sum_{i=1}^n\left(\tilde w(v)^\top z_i\right)^2}.
$$

By Lemma C.3, for any $\epsilon\in(0,1)$, select $\delta=\epsilon^d$, if we choose $n\gtrsim\left(d\log\left(1/\epsilon\right)+\log(1/\delta)\right)/\epsilon^2\gtrsim d\log\left(1/\epsilon\right)/\epsilon^2$, then w.p. at least $1-\delta=1-\epsilon^d$, the following inequalities hold:

$$
\sup_{\theta\in\mathbb{S}^{d-1}}\left|\frac{1}{n}\sum_{i=1}^n\left(\tilde\theta(v)^\top z_i\right)^2-1\right|\leq\epsilon,\quad\sup_{v\in\mathbb{S}^{d-1}}\left|\frac{1}{n}\sum_{i=1}^n\left(\tilde w(v)^\top z_i\right)^2-1\right|\leq\epsilon.
$$

Then we consider the estimate about the numerator of $g(\theta, v)$ for $K$ fixed direction $v \in \mathcal{V} = \{v_1, \cdots, v_K\}$. By Lemma C.4, for any $0 < \epsilon < 1/2$, there are constants $C_1 = C_1(\epsilon) > 0$ and $C_2 = C_2(\epsilon) > 0$, such that if $n \geq C_1 d \log d$, then with probability at least $1 - \frac{C_2}{n^2}$, it holds

$$
\begin{aligned}
&\mathbb{P}\left(\sup_{v \in \mathcal{V}} \sup_{\theta \in \mathbb{S}^{d-1}} \left| \frac{1}{n} \sum_{i=1}^{n} \left(\tilde{r}(\theta)^\top z_i\right)^2 \left(\tilde{w}(v)^\top z_i\right)^2 - \mathbb{E}\left[\left(\tilde{r}(\theta)^\top z_1\right)^2 \left(\tilde{w}(v)^\top z_1\right)^2\right] \right| \geq \epsilon \right) \\
&\leq \sum_{k=1}^{K} \mathbb{P}\left(\sup_{\theta \in \mathbb{S}^{d-1}} \left| \frac{1}{n} \sum_{i=1}^{n} \left(\tilde{r}(\theta)^\top z_i\right)^2 \left(\tilde{w}(v_k)^\top z_i\right)^2 - \mathbb{E}\left[\left(\tilde{r}(\theta)^\top z_1\right)^2 \left(\tilde{w}(v_k)^\top z_1\right)^2\right] \right| \geq \epsilon \right) \\
&\leq \frac{K C_2}{n^2}.
\end{aligned}
$$

Now we combine these two bounds. For any $\epsilon \in (0, 1/2)$, there exist $\tilde{C}_1(\epsilon) := \max\{C(\epsilon), \log(1/\epsilon)/\epsilon^2\} > 0$ and $C_2 = C_2(\epsilon) > 0$, such that: if we choose $n \gtrsim \tilde{C}_1 d \log d$, then $w.p.$ at least $1 - \frac{C_2}{n^2} - \epsilon^d$,

$$
\sup_{\theta \in \mathbb{S}^{d-1}} \left| \frac{1}{n} \sum_{i=1}^{n} \left(\tilde{\theta}(v)^\top z_i\right)^2 - 1 \right| \leq \epsilon, \qquad \sup_{v \in \mathbb{S}^{d-1}} \left| \frac{1}{n} \sum_{i=1}^{n} \left(\tilde{w}(v)^\top z_i\right)^2 - 1 \right| \leq \epsilon,
$$

$$
\sup_{v \in \mathcal{V}} \sup_{\theta \in \mathbb{S}^{d-1}} \left| \frac{1}{n} \sum_{i=1}^{n} \left(\tilde{r}(\theta)^\top z_i\right)^2 \left(\tilde{w}(v)^\top z_i\right)^2 - \mathbb{E}\left[\left(\tilde{r}(\theta)^\top z_1\right)^2 \left(\tilde{w}(v)^\top z_1\right)^2\right] \right| \geq \epsilon.
$$

Therefore, for any $\theta \in \mathbb{R}^p$ and $v \in \mathcal{V}$,

$$
\frac{\mathbb{E}\left[(\tilde{r}(\theta)^\top z_1)^2 (\tilde{w}(v)^\top z_1)^2\right] - \epsilon}{(1 + \epsilon)^2} \leq g(\theta; v) \leq \frac{\mathbb{E}\left[(\tilde{r}(\theta)^\top z_1)^2 (\tilde{w}(v)^\top z_1)^2\right] + \epsilon}{(1 - \epsilon)^2}.
$$

Then in the same way as the proof of Lemma 4.2, it holds that

$$
\frac{1 - \epsilon}{(1 + \epsilon)^2} \leq g(\theta; v) \leq \frac{3 + \epsilon}{(1 - \epsilon)^2}.
$$

Thus, we complete the proof.

$\square$

# D  Proofs in Section 5.1: Convergence rate of SGD

## D.1  Proof of Theorem 5.2

For linear regression, it holds that:

$$
\mathcal{L}(\theta_{t+1}) = \mathcal{L}(\theta_t - \eta \nabla \ell_{\xi_t}(\theta_t)) = \mathcal{L}(\theta_t) - \eta \left\langle \nabla \mathcal{L}(\theta_t), \nabla \ell_{\xi_t}(\theta_t) \right\rangle + \frac{\eta^2}{2} \nabla \ell_{\xi_t}^\top(\theta_t) G \nabla \ell_{\xi_t}(\theta_t).
$$

Then taking the expectation, we have:

$$
\begin{aligned}
\mathbb{E}\left[\mathcal{L}(\theta_{t+1})\right] &= \mathbb{E}\left[\mathcal{L}(\theta_t)\right] - \eta \mathbb{E}\left[\|\nabla \mathcal{L}(\theta_t)\|^2\right] + \frac{\eta^2}{2} \mathbb{E}\left[\mathrm{tr}\left(\nabla \ell_{\xi_t}^\top(\theta_t) G \nabla \ell_{\xi_t}(\theta_t)\right)\right] \\
&= \mathbb{E}\left[\mathcal{L}(\theta_t)\right] - \eta \mathbb{E}\left[\|\nabla \mathcal{L}(\theta_t)\|^2\right] + \frac{\eta^2}{2} \mathbb{E}\left[\mathrm{tr}\left(\Sigma(\theta_t) G\right)\right] \\
&= \mathbb{E}\left[\mathcal{L}(\theta_t)\right] - \eta \mathbb{E}\left[\|\nabla \mathcal{L}(\theta_t)\|^2\right] + \frac{\eta^2 \|G\|_F^2}{2} \mathbb{E}\left[\mu(\theta_t) \mathcal{L}(\theta_t)\right]
\end{aligned}
$$

$$\overset{\mu(\theta)\leq 3.1}{\leq} \mathbb{E}\left[\mathcal{L}(\theta_t)\right] - \eta\mathbb{E}\left[\|\nabla\mathcal{L}(\theta_t)\|^2\right] + \frac{3.1\eta^2\|G\|_F^2}{2}\mathbb{E}\left[\mathcal{L}(\theta_t)\right]$$

$$\overset{\text{strongly convex}}{\leq} \left(1 - 2\eta\lambda_{\min}(G) + \frac{3.1\eta^2\|G\|_F^2}{2}\right)\mathbb{E}\left[\mathcal{L}(\theta_t)\right]$$

$$\overset{\eta=\eta^*=\frac{\lambda_{\min}(G)}{2\|G\|_F^2}}{\leq} \left(1 - \frac{\lambda_{\min}^2(G)}{2\|G\|_F^2}\right)\mathbb{E}\left[\mathcal{L}(\theta_t)\right].$$

Thus, we obtain the convergence rate:

$$\mathbb{E}\left[\mathcal{L}(\theta_t)\right] \leq \left(1 - \frac{\lambda_{\min}^2(G)}{2\|G\|_F^2}\right)^t \mathbb{E}\left[\mathcal{L}(\theta_0)\right].$$

# E Proofs in Section 5.2: Escape direction of SGD

## E.1 Proof of Theorem 5.4

Recall that $w(t) = \sum_{i=1}^d w_i(t)u_i$ with $w_i(t) = u_i^\top w(t)$. Then, $w_i(t+1) = (1 - \eta\lambda_i)w_i(t) + \eta\xi(t)^\top u_i$. Taking the expectation of the square of both sides, we obtain

$$\mathbb{E}\left[w_i^2(t+1)\right] = (1 - \eta\lambda_i)^2\mathbb{E}\left[w_i^2(t)\right] + \eta^2\mathbb{E}[|u_i^\top\xi(t)|^2],$$

According to Assumption 5.3, there exists $A_1, A_2 > 0$ such that for any $i \in [d]$,

$$A_1\lambda_i\mathcal{L}(w_t) \leq \mathbb{E}[|u_i^T\xi(t)|] \leq A_2\lambda_i\mathcal{L}(w_t).$$

Let $X_t = \sum_{i=1}^k \lambda_i\mathbb{E}[w_i^2(t)], Y_t = \sum_{i=k+1}^d \lambda_i\mathbb{E}[w_i^2(t)]$ denote the components of loss energy along sharp and flat directions, respectively. And we denote $D_k(t) := Y_t/X_t$.

Plugging the fact that $2\mathcal{L}(w(t)) = X_t + Y_t$ into the two formulations above, we can obtain the following component dynamics:

$$X_{t+1} \leq \alpha_k X_t + A_2\eta^2(\sum_{i=1}^k \lambda_i^2)(X_t + Y_t),$$

$$X_{t+1} \geq A_1\eta^2(\sum_{i=1}^k \lambda_i^2)(X_t + Y_t), \tag{12}$$

$$Y_{t+1} \geq A_1\eta^2\big(\sum_{i=k+1}^d \lambda_i^2\big)(X_t + Y_t),$$

where $\alpha_k \leq \max_{i=1,\ldots,k}|1 - \eta\lambda_i|^2$. The terms $\alpha_k X_t$ and $\beta_k Y_t$ capture the impact of the gradient, while the remaining terms originate from the noise.

From (12), we have the following estimate about $D_k(t+1)$:

$$D_k(t+1) = \frac{Y_{t+1}}{X_{t+1}} \geq \frac{A_1\eta^2\big(\sum_{i=k+1}^d \lambda_i^2\big)(X_t + Y_t)}{\alpha_k X_t + A_2\eta^2(\sum_{i=1}^k \lambda_i^2)(X_t + Y_t)}$$

$$= \frac{A_1\sum_{i=k+1}^d \lambda_i^2}{A_2\sum_{i=1}^k \lambda_i^2} \cdot \frac{1}{1 + \frac{\alpha_k}{A_2\eta^2\sum_{i=1}^k \lambda_i^2}\frac{X_t}{X_t+Y_t}} \tag{13}$$

$$\geq \frac{A_1\sum_{i=k+1}^d \lambda_i^2}{A_2\sum_{i=1}^k \lambda_i^2} \cdot \frac{1}{1 + \frac{\max\limits_{1\leq i\leq k}|1-\eta\lambda_i|^2}{A_2\eta^2\sum_{i=1}^k \lambda_i^2}\frac{X_t}{X_t+Y_t}}.$$

We will prove this theorem for the learning rate $\eta = \frac{\beta}{\|G(\theta^*)\|_F}$, where $\beta \geq \frac{1.1}{\sqrt{A_1}}$.

**Case (I). Small learning rate** $\eta \in [\frac{1.1}{\sqrt{A_1}\|G(\theta^*)\|_{\mathrm{F}}}, \frac{1}{\lambda_1}]$. In this step, we consider $\eta = \frac{\beta}{\|G(\theta^*)\|_{\mathrm{F}}}$ such that $\beta \geq \frac{1.1}{\sqrt{A_1}}$ and $\eta \leq \frac{1}{\lambda_1}$. Then we have:

$$\frac{\max\limits_{1 \leq i \leq k} |1 - \eta\lambda_i|^2}{A_2\eta^2 \sum_{i=1}^{k} \lambda_i^2} \leq \frac{1}{A_2\eta^2 \sum_{i=1}^{k} \lambda_i^2}.$$

Notice that (12) also ensures:

$$(X_{t+1} + Y_{t+1}) \geq A_1\eta^2 \Big( \sum_{i=1}^{d} \lambda_i^2 \Big)(X_t + Y_t).$$

Combining this inequality with (12), we have the estimate:

$$\frac{X_{t+1}}{X_{t+1} + Y_{t+1}} \leq \frac{\alpha_k X_t + A_2\eta^2(\sum_{i=1}^{k} \lambda_i^2)(X_t + Y_t)}{X_{t+1} + Y_{t+1}}$$

$$\leq \frac{\alpha_k X_t}{A_1\eta^2 \big( \sum_{i=1}^{d} \lambda_i^2 \big)(X_t + Y_t)} + \frac{A_2(\sum_{i=1}^{k} \lambda_i^2)}{A_1 \big( \sum_{i=1}^{d} \lambda_i^2 \big)}$$

For simplicity, we denote $W_t := \frac{X_t}{X_t + Y_t}$, $A := \frac{\alpha_k}{A_1\eta^2 \big( \sum_{i=1}^{d} \lambda_i^2 \big)}$, and $B := \frac{A_2(\sum_{i=1}^{k} \lambda_i^2)}{A_1 \big( \sum_{i=1}^{d} \lambda_i^2 \big)}$.

From $\eta \leq 1/3$, we have $\alpha_k \leq 1$ and $A \leq \frac{1}{A_1\eta^2 \big( \sum_{i=1}^{d} \lambda_i^2 \big)} = \frac{1}{A_1\beta^2} < 1$. Moreover, it holds that

$$W_{t+1} \leq AW_t + B \leq A(AW_{t-1} + B) + B = A^2 W_{t-1} + B(1 + A)$$

$$\leq \cdots \leq A^{t+1} W_0 + B(1 + A + \cdots + A^t) = A^{t+1} W_0 + \frac{1 - A^{t+1}}{1 - A} B$$

On the one hand, if we choose

$$t \geq \frac{\log \Big( 1/W_0 A_2\eta^2 \sum_{i=1}^{k} \lambda_i^2 \Big)}{\log (A_1\beta^2)},$$

then we have

$$A^t W_0 \leq \left( \frac{\alpha_k}{A_1\eta^2(\sum_{i=1}^{d} \lambda_i^2)} \right)^t W_0 \leq \left( \frac{1}{A_1\beta^2} \right)^t W_0 \leq A_2\eta^2 \sum_{i=1}^{k} \lambda_i^2.$$

On the other hand, if we choose $t \geq 1$, then it holds that

$$\frac{1 - A^t}{1 - A} B \leq B = \frac{A_2(\sum_{i=1}^{k} \lambda_i^2)}{A_1 \big( \sum_{i=1}^{d} \lambda_i^2 \big)} \leq A_2\eta^2 \sum_{i=1}^{k} \lambda_i^2.$$

Hence, if we choose

$$t \geq \max \left\{ 1, \frac{\log \Big( 1/W_0 A_2\eta^2 \sum_{i=1}^{k} \lambda_i^2 \Big)}{\log (A_1\beta^2)} \right\},$$

then we have

$$\frac{X_t}{X_t + Y_t} = W_t \leq A^t W_0 + \frac{1 - A^t}{1 - A} B \leq 2A_2\eta^2 \sum_{i=1}^{k} \lambda_i^2,$$

which implies that

$$\text{RHS of (13)} \geq \frac{A_1 \sum_{i=k+1}^d \lambda_i^2}{A_2 \sum_{i=1}^k \lambda_i^2} \cdot \frac{1}{1 + \frac{\max\limits_{1 \leq i \leq k} |1-\eta\lambda_i|^2}{A_2 \eta^2 \sum_{i=1}^k \lambda_i^2} \frac{X_t}{X_t+Y_t}}$$

$$\geq \frac{A_1 \sum_{i=k+1}^d \lambda_i^2}{A_2 \sum_{i=1}^k \lambda_i^2} \cdot \frac{1}{1 + \frac{1}{A_2 \eta^2 \sum_{i=1}^k \lambda_i^2} \cdot 2A_2\eta^2 \sum_{i=1}^k \lambda_i^2} = \frac{A_1 \sum_{i=k+1}^d \lambda_i^2}{3A_2 \sum_{i=1}^k \lambda_i^2}.$$

**Case (II). Large learning rate $\eta \geq 1/\lambda_1$.** In this step, we consider $\eta \geq \frac{1}{\lambda_1}$. Then for any $t \geq 0$, we have:

$$\text{RHS of (13)} = \frac{A_1 \sum_{i=k+1}^d \lambda_i^2}{A_2 \sum_{i=1}^k \lambda_i^2} \cdot \frac{1}{1 + \frac{\alpha_k}{\sum_{i=k+1}^d \lambda_i^2} \frac{X_t}{X_t+Y_t}} \geq \frac{A_1 \sum_{i=k+1}^d \lambda_i^2}{A_2 \sum_{i=1}^k \lambda_i^2} \cdot \frac{1}{1 + \frac{\max\limits_{i \in [k]} |1-\eta\lambda_i|^2}{A_2 \eta^2 \sum_{i=1}^k \lambda_i^2}}$$

$$\geq \frac{A_1 \sum_{i=k+1}^d \lambda_i^2}{A_2 \sum_{i=1}^k \lambda_i^2} \cdot \frac{1}{1 + \frac{\max\{1, |1-\eta\lambda_1|^2\}}{A_2 \eta^2 \sum_{i=1}^k \lambda_i^2}} \geq \frac{A_1 \sum_{i=k+1}^d \lambda_i^2}{A_2 \sum_{i=1}^k \lambda_i^2} \cdot \frac{1}{1 + \frac{1}{A_2}} = \frac{A_1 \sum_{i=k+1}^d \lambda_i^2}{(A_2+1) \sum_{i=1}^k \lambda_i^2}.$$

Combining Case (I) and (II), we obtain this theorem: If we choose the learning rate $\eta = \frac{\beta}{\|G(\theta)\|_F}$, where $\beta \geq \frac{1.1}{\sqrt{A_1}}$, then for any

$$t \geq \max \left\{ 1, \frac{\log\left(1/W_0 A_2 \eta^2 \sum_{i=1}^k \lambda_i^2\right)}{\log\left(A_1 \beta^2\right)} \right\},$$

we have

$$D_k(t+1) \geq \frac{A_1 \sum_{i=k+1}^d \lambda_i^2}{\max\{3A_2, A_2+1\} \sum_{i=1}^k \lambda_i^2}.$$

$\square$

## E.2 Proof of Proposition 5.5

Recall that $w(t) = \sum_{i=1}^d w_i(t) u_i$ with $w_i(t) = u_i^\top w(t)$. Then, for GD, $w_i(t+1) = (1 - \eta\lambda_i) w_i(t)$, which implies:

$$w_i(t) = (1 - \eta\lambda_i)^t w_i(0).$$

Therefore, for $\eta = \beta/\lambda_1$ ($\beta > 2$), it holds that

$$D_1(t) = \frac{\sum_{i=2}^d \lambda_i w_i^2(t)}{\lambda_1 w_1^2(t)} = \frac{\sum_{i=2}^d \lambda_i (1-\eta\lambda_i)^{2t} w_i^2(0)}{\lambda_1 (1-\eta\lambda_1)^{2t} w_1^2(0)}.$$

$\square$

# F Useful Inequalities

**Lemma F.1** (Bernstein's Inequality (Vershynin, 2018))**.** *Suppose $\{X_1, \cdots, X_n\}$ are independent sub-Exponential random variables with $\|X_i\|_{\psi_1} \leq K$. Then there exists an absolute constant $c > 0$ such that for any $t \geq 0$, we have:*

$$\mathbb{P}\left( \left| \frac{1}{n} \sum_{i=1}^n X_i - \frac{1}{n} \sum_{i=1}^n \mathbb{E}[X_i] \right| > t \right) \leq 2 \exp\left( -cn \min\left\{ \frac{t}{K}, \frac{t^2}{K^2} \right\} \right).$$

**Lemma F.2** (Hanson-Wright's Inequality (Vershynin, 2018))**.** *Let $X = (X_1, \cdots, X_n) \in \mathbb{R}^n$ be a random vector with independent mean zero sub-Gaussian coordinates. Let $A$ be an $n \times n$ matrix. Then, there exists an absolute constant $c$ such that for every $t \geq 0$, we have*

$$\mathbb{P}\left(\left|X^\top AX - \mathbb{E}[X^\top AX]\right| \geq t\right) \leq 2 \exp\left(-c \min\left\{\frac{t^2}{K^4 \|A\|_{\mathrm{F}}^2}, \frac{t}{K^2 \|A\|_2}\right\}\right),$$

*where $K = \max_i \|X_i\|_{\psi_2}$.*

**Lemma F.3** (Covariance Estimate for sub-Gaussian Distribution (Vershynin, 2018))**.** *Let $x, x_1, \cdots, x_n$ be i.i.d. random vectors in $\mathbb{R}^d$. More precisely, assume that there exists $K \geq 1$ s.t. $\|\langle x, v \rangle\|_{\psi_2} \leq K \|\langle x, v \rangle\|_{L_2}$ for any $v \in \mathbb{S}^{d-1}$, Then for any $u \geq 0$, w.p. at least $1 - 2\exp(-u)$ one has*

$$\left\|\frac{1}{n}\sum_{i=1}^n x_i x_i^\top - \mathbb{E}[xx^\top]\right\| \leq CK^2 \left(\sqrt{\frac{d+u}{n}} + \frac{d+u}{n}\right) \left\|\mathbb{E}[xx^\top]\right\|,$$

*where $C$ is an absolute positive constant.*

**Definition F.4** (Sub-Weibull Distribution)**.** We define $X$ as a sub-Weibull random variable if it has a bounded $\psi_\beta$-norm. The $\psi_\beta$-norm of $X$ for any $\beta > 0$ is defined as

$$\|X\|_{\psi_\beta} := \inf\left\{C > 0 : \mathbb{E}\left[\exp(|X|^\beta/C^\beta)\right] \leq 2\right\}.$$

Particularly, when $\beta = 1$ or $2$, sub-Weibull random variables reduce to sub-Exponential or sub-Gaussian random variables, respectively.

**Lemma F.5** (Concentration Inequality for Sub-Weibull Distribution, Theorem 3.1 in (Hao et al., 2019))**.** *Suppose $\{X_i\}_{i=1}^n$ are independent sub-Weibull random variables with $\|X_i\|_{\psi_\beta} \leq K$. Then there exists an absolute constant $C_\beta$ only depending on $\beta$ such that for any $\delta \in (0, 1/e^2)$, w.p. at least $1 - \delta$, we have*

$$\left|\frac{1}{n}\sum_{i=1}^n X_i - \frac{1}{n}\sum_{i=1}^n \mathbb{E}[X_i]\right| \leq C_\beta K\left(\left(\frac{\log(1/\delta)}{n}\right)^{1/2} + \frac{\left(\log(1/\delta)\right)^{1/\beta}}{n}\right).$$

**Lemma F.6** (Cauchy-Schwarz Inequalities)**.**
*(1) Let $S \in \mathbb{R}^{n \times n}$ be a positive symmetric definite matrix. For any $x, y \in \mathbb{R}^n$, we denote $\langle x, y \rangle_S := x^\top Sy$ and $\|x\|_S := \sqrt{\langle x, x \rangle_S}$, then we have $|\langle x, y \rangle_S| \leq \|x\|_S \|y\|_S$.*
*(2) Given two random variables $X$ and $Y$, it holds that $|\mathbb{E}[XY]| \leq \sqrt{\mathbb{E}[X^2]}\sqrt{\mathbb{E}[Y^2]}$.*

