# OpenReview forum: "The Noise Geometry of Stochastic Gradient Descent"
_TMLR — Rejected by TMLR_

### Review · Reviewer_pLSN · 2025-05-26

**Summary Of Contributions:**

This paper investigates the noise geometry of Stochastic Gradient Descent (SGD), specifically how the heterogeneous structure of minibatch noise aligns favorably with the local geometry of the loss landscape. The main contributions include proposing a novel metric $\mu(\theta)$ to quantitatively characterize this alignment. The authors theoretically demonstrate that $\mu(\theta)$ is close to 1 across the parameter space for over-parameterized linear models and two-layer nonlinear networks, and similar directional alignment holds for linear models. Leveraging these characterizations, the paper provides a refined convergence rate analysis for SGD in high-dimensional linear regression and a detailed examination of how SGD escapes sharp minima, showing that SGD tends to escape along flatter directions, unlike GD which escapes along the sharpest directions. Experimental results on synthetic and real-world datasets (e.g., CIFAR-10 with VGG and ResNets) support their theoretical findings regarding noise alignment and escape directions.

**Audience:**

Yes

**Broader Impact Concerns:**

This paper features a theoretical study of the model training. Given its foundational nature, direct negative societal impacts are highly unlikely.

**Claims And Evidence:**

Yes

**Requested Changes:**

1. How do we derive the expression (6)?

2. In the definition of OLMs, is there any requirement for the relation between $d$ and $p$?

3. It is suggested to use consistent notations in different examples in “Over-parameterized linear models (OLMs)” in page 4.

4. If I did not miss anything, $\bar\gamma_1(\theta) = \bar\gamma(\theta):=2\mathcal{L}(\theta)\|G(\theta)\|^2_F$, while $\gamma_1(\theta)=\gamma(\theta)-\nabla\mathcal{L}(\theta)^\top H(\theta)\nabla\mathcal{L}(\theta)\ne\gamma(\theta)$ . However, the authors are trying to show both $\gamma(\theta)=\bar\gamma(\theta)$ (as mentioned in Sec 1.1) and  $\gamma_1(\theta)=\bar\gamma_1(\theta)$ (as mentioned in the beginning of Sec 3). Please clarify which one is correct. Generally, given $\nabla\mathcal{L}(\theta)^\top H(\theta)\nabla\mathcal{L}(\theta)\ne 0$, does Theorem 3.2 still hold of $\frac{\gamma(\theta)}{\bar\gamma(\theta)}$?

5. The paper claims that the approximation (4) “is neither theoretically justified nor empirically verified.” Is it possible to provide any empirical evidence that (4) does not necessarily hold in the practical setting?

**Strengths And Weaknesses:**

Strengths:

- The introduction of $\mu(\theta)$ provides concrete, quantifiable ways to measure the alignment between SGD noise and the loss landscape, moving beyond heuristic approximations.

- The paper offers rigorous theoretical proofs for the proposed alignment properties in various model settings (over-parameterized linear models, two-layer networks), showing the alignment holds regardless of over-parameterization.

- The refined convergence rate for SGD in high-dimensional linear regression, explicitly leveraging the structured nature of noise, is a significant theoretical advancement compared to prior work that overlooked noise shape.

- The detailed analysis of SGD's escape direction from sharp minima, demonstrating its preference for flatter directions, provides valuable insights into SGD's implicit regularization and generalization properties.

- The theoretical findings are well-supported by both small-scale synthetic experiments and larger-scale real-world experiments on deep neural networks (VGG, ResNet on CIFAR-10).


Weaknesses:

- Assumptions 5.3 may be able to derive by extending the technique in this paper.

- The rigorous proof only holds under the regression tasks, and the model is linear with respect to x.

- The “large-scale” experiments are in the order of millions parameter level, which is only a moderate scale.

---

### Review · Reviewer_rFet · 2025-06-15

**Summary Of Contributions:**

In this paper, the authors perform an extensive analysis of the noise geometry of SGD. The authors present some metrics that describe the effect of noise on the loss behavior and subspace projection dynamics. To validate the theoretical calculations, experiments were also performed.

**Audience:**

Yes

**Broader Impact Concerns:**

I do not have any concerns on the ethical implications of this paper.

**Claims And Evidence:**

Yes

**Requested Changes:**

To improve the readability of the work, I would suggest a few points.

$\bullet$ I suggest to double-check the size of brackets in the formulas. For example, in Theorem 5.2, page 7; in the paragraph "However, as explained above...", page 8.

$\bullet$ It is worth considering to relocate section "Useful Inequalities" in the start of the Appendix part. Then the proof section would be understood more easily since the introductions for comprehension would have been given earlier.

**Strengths And Weaknesses:**

Theoretical analysis can be emphasized as a serious plus. SGD is still being researched today, despite the fact that the method itself emerged in the previous century. The results presented by the authors provide an even better understanding of how SGD works in practice in terms of stochasticity itself and its effect on the loss function landscape.

The clean writing of the text is also worth noting:
1) The text is clearly written.
2) The Related Work section is quite thorough in terms of reviewing the existing literature.

Regarding to the proof -- I've checked the Appendix part. I have not found any technical flaws.

Fortunately, there are no major flaws that I can point out in this study. Regarding the shortcomings -- see **Requested Changes**.

---

### Review · Reviewer_S8AS · 2025-06-19

**Summary Of Contributions:**

This paper studies the geometry of the noise induced by stochastic gradient descent (SGD). The main idea is to quantify how the structure of stochastic noise aligns with the curvature of the loss landscape. The authors use two metrics to capture this alignment (see Def. 3.1 and Def. 4.1) and study these mainly in over-parameterized linear models and two-layer neural networks. They argue that this alignment has implications for the convergence of SGD and for how SGD escapes sharp minima, showing that SGD tends to escape along flatter directions compared to standard gradient descent.

**Audience:**

Yes

**Claims And Evidence:**

Yes

**Requested Changes:**

See weaknesses above.

**Strengths And Weaknesses:**

**Strengths**
- The paper touches on an important property of SGD, which is central in modern optimization. Understanding these dynamics can potentially help us in better optimizer design.

**Weaknesses**
- The paper is hard to follow. The presentation, structure and notation are messy, making it difficult to read and understand the main contributions.
- The scope is very limited: only over-parameterized linear models and 2-layer neural networks are considered, so the results feel narrow and not very general.
- Although the title and abstract mention mini-batch SGD, the paper actually focuses only on the vanilla SGD case (batch size B=1), which is misleading. It would be much more interesting to see the effects of larger mini-batches.
- It seems that only a constant learning rate is considered. The effects of more realistic, decaying or adaptive learning rates are not addressed.
- The variance assumptions are quite strict. The authors should study more general and modern assumptions, such as quadratic growth or the ABC assumption.

---

### Decision · Action_Editor_Uc4K · 2025-07-29

**Recommendation:** Reject

**Additional Comments:**

While the reviewers were positive in the review step, important points were raised. Unfortunately, the authors did not respond to any of the concerns raised and did not revise their paper to address the comments. For this reason, I believe the paper needs to undergo substantial revision to correct all major and minor issues pointed out in the review stage. I highly recommend that the authors resubmit their paper to TMLR. I believe it is a valuable contribution, but it cannot be accepted in its current form

**Audience:**

Yes

**Audience Explanation:**

Stochastic gradient descent (SGD) and its variants are widely used in machine learning. Understanding the dynamics of SGD is an important problem that holds significant value for the community. The findings in this paper have implications for generalization, which is a crucial property of the optimizer.

**Claims And Evidence:**

No

**Claims Explanation:**

The authors study the relation between the noise induced by stochastic gradient descent and its alignment with the landscape of the loss. They mainly focus on overparametrized two-layer linear networks and present two metrics to quantify the studied alignment.  They present convincing evidence that the noise affects how the optimizer (SGD) escapes sharp minima.  They demonstrate that, unlike gradient descent, which escapes sharp regions along the steepest directions, SGD tends to escape through flatter areas. Some of the results are clear and convincing as they support their claims with both synthetic and real data experiments. However, as pointed out by several reviewers, the paper includes some flaws that need to be addressed. Unfortunately, the authors did not engage in the rebuttal and did not correct these flaws. Here are a few examples raised by the reviewers:

-The results are restricted to linear two-layer networks.

-The analysis does not generalize to broader model classes or more realistic settings.

-Despite mentioning "mini-batch SGD" in the title and abstract, the analysis focuses only on batch size = 1.

-This discrepancy was noted as misleading and reduces practical relevance.

-Simplistic assumptions.

-Several requested changes.

**Resubmission Of Major Revision:**

The authors may consider submitting a major revision at a later time.